# Vector Memory and Role-Conditioned Multi-Agent Systems: Two Extensions to Improve Reflexion for Language Model Self-Improvement

## Abstract

REFLEXION improves language model performance through verbal self-reflection, but two design choices limit its reach. Its memory is a recency-ordered sliding window that evicts old reflections as new ones arrive, regardless of which are actually relevant. Moreover, a single model simultaneously generates a solution, critiques it, and plans the next attempt, roles that genuinely benefit from separation. In this paper, we address both of these limitations. We replace the sliding window with vector episodic memory, which stores Sentence-BERT embeddings alongside each reflection and retrieves them based on cosine similarity rather than recency. We also split the single Agent into a Generator, a Critic, and a Verifier, each of which draws on a shared role-conditioned memory pool. The results are quite encouraging. In a controlled benchmark in which 9 distractor tasks bury the relevant memories, temporal memory fails at the task level (0% Session-5 task success; dependency recall is 50%, reflecting partial buffer retention that is nonetheless insufficient when both Session-1 learnings are required simultaneously, see Remark 5.1). Vector memory succeeds in all 3 independent trials on this diagnostic setup (100% Session-5 task success), at a constant ~14 ms overhead that stays flat up to 50,000 stored reflections. This result comes from a controlled synthetic benchmark specifically designed to stress-test FIFO eviction; on the natural HumanEval coding benchmark, the gain is more modest ($+3.7$ pp, $p = 0.033$, $d = 0.127$), consistent with the independence of HumanEval tasks. On 164 HumanEval coding tasks with Google Gemini 2.5 Flash, the vector extension reaches Pass@3 = 92.7% ($+3.7$ pp, $p = 0.033$, $d = 0.127$), and the multi-agent system reaches Pass@3 = 96.3% ($+7.9$ pp, $p < 0.001$, $d = 0.301$) with Pass@1 = 93.9% ($+12.2$ pp over the modular baseline), a first-attempt gain that predates any reflection at all. Our results suggest that semantic retrieval is particularly important when tasks are correlated across sessions, while role separation provides the greatest benefit on independent code-generation tasks.

## 1 Introduction

Large language models (LLMs) have evolved from passive text generators into systems capable of reasoning, code synthesis, and multi-step decision-making across extended interactions. Recent work has demonstrated their ability to act as autonomous agents in complex environments, including program synthesis, embodied reasoning, and interactive decision support (Pang et al., 2024; Acikgoz et al., 2025; Hu et al., 2025). As these systems are deployed in increasingly long-horizon and multi-task settings, a central challenge is enabling them to improve from experience without relying on expensive parameter updates.

A promising paradigm addressing this challenge is verbal reinforcement learning (VRL), in which agents iteratively refine their behavior using natural-language feedback derived from prior failures, rather than through gradient-based updates to model parameters. In VRL, agents maintain an external memory of self-generated reflections that condition future actions. A prominent example of this approach is Reflexion (Shinn et al., 2023), which demonstrates that agents can improve performance by storing and reusing natural language self-critiques across trials. More broadly, VRL-style methods have shown strong empirical

performance across tasks such as code generation and sequential decision-making (Pang et al., 2024; Acikgoz et al., 2025). This approach is attractive due to its model-agnostic nature, interpretability, and low computational cost. However, recent studies suggest that the effectiveness of such self-improvement mechanisms depends critically on how memory is structured and how feedback is generated and utilized (Du, 2026; Lu et al., 2026; Ma et al., 2026).

Despite strong empirical performance, existing VRL frameworks, most notably Reflexion-style approaches (Shinn et al., 2023), embed architectural assumptions that limit their applicability in realistic deployment scenarios. One key limitation of episodic memory lies in its design: many systems rely on fixed-size, recency-based buffers, which implicitly assume that recently observed information is most relevant. However, prior work has shown that LLMs struggle to utilize long contexts effectively and may ignore relevant information when it is not positioned favorably (Liu et al., 2024). In long-horizon or multi-session settings, this can cause semantically important reflections to be overwritten by unrelated experiences, resulting in degraded performance. Recent research on agent memory systems emphasizes the need for structured, retrieval-based memory mechanisms that prioritize semantic relevance over temporal proximity (Xu et al., 2025; Lu et al., 2026; Du, 2026).

A second limitation arises from the use of a single model to perform generation, evaluation, and self-reflection, as in Reflexion (Shinn et al., 2023). These operations require distinct reasoning modes: generation benefits from confident hypothesis construction, whereas evaluation requires critical and adversarial reasoning. Prior work has shown that LLMs exhibit limited ability to self-correct their own outputs, often reinforcing initial errors rather than identifying them (Huang et al., 2024; Saadat & Nemzer, 2026). This tendency is linked to correlated error patterns and confirmation bias in autoregressive generation, where the model's initial output influences subsequent reasoning, making it less likely to identify its own mistakes. Recent multi-agent approaches address this limitation by decomposing reasoning into specialized roles, enabling more robust and diverse forms of reasoning (Motwani et al., 2025; Chen et al., 2025; Khan et al., 2025).

Motivated by these limitations, we revisit the design of memory and reasoning in VRL-based agents, building directly on the Reflexion framework (Shinn et al., 2023), and propose two complementary architectural modifications. First, we replace recency-based memory with a semantic retrieval mechanism, *VectorEpisodicMemory*, which stores reflections as dense embeddings and retrieves them based on similarity to the current task. This design aligns with emerging trends in agent memory research that advocate embedding-based retrieval and structured memory organization for long-term reasoning (Xu et al., 2025; Ma et al., 2026; Du, 2026). Second, we introduce a role-separated multi-agent pipeline consisting of a Generator, Critic, and Verifier, where each component is responsible for a distinct stage of reasoning. This decomposition enforces separation between generation and evaluation, improving the quality of feedback and reducing self-confirmation bias, in line with recent advances in multi-agent LLM systems (Motwani et al., 2025; Samanta et al., 2026; Härer, 2025).

We show that these modifications target distinct but complementary failure modes: semantic memory retrieval is highly beneficial in long-horizon settings where relevant information must persist across temporally distant interactions. At the same time, role separation improves within-trial reasoning by enabling more effective critique and correction. Together, these results highlight the importance of architectural design choices in VRL systems and offer guidance for selecting mechanisms suited to specific deployment scenarios.

We address both limitations and make the following contributions:

1. We identify a failure mode of recency-based memory in Reflexion-style agents, showing that fixed-size FIFO buffers can systematically discard semantically relevant experience in long-horizon settings.

2. We propose *Vector Episodic Memory*, a drop-in replacement for recency-based buffers that retrieves reflections using embedding-based similarity, enabling robust reuse of past experience across tasks.

3. We introduce a multi-agent Reflexion architecture with Generator–Critic–Verifier role separation, improving the quality of self-evaluation and first-attempt performance.

4. We provide empirical evidence across long-horizon and single-trial settings demonstrating that semantic retrieval and role separation yield consistent performance improvements.

The central lesson is that the optimal architectural choice depends on the deployment setting: semantic memory retrieval is critical when tasks are temporally correlated. At the same time, multi-agent role separation provides the greatest benefit for improving single-trial reasoning in our evaluated setting.

## 2 Related Work

### 2.1 Reflexion and Iterative Self-Improvement

REFLEXION (Shinn et al., 2023) extends chain-of-thought prompting (Wei et al., 2022) and self-consistency (Wang et al., 2023) to multi-trial settings by storing verbal self-critiques in episodic memory. The framework operates as an Actor–Evaluator–Self-Reflection loop: a language agent attempts a task, receives binary or scalar feedback from an evaluator, and generates a natural language reflection that is prepended to the next trial's context. This verbal reinforcement signal avoids gradient updates entirely, making the framework model-agnostic and applicable to any black-box LLM. Shinn et al. (Shinn et al., 2023) demonstrate strong results on HumanEval (Chen et al., 2021), AlfWorld (Shridhar et al., 2021), and HotpotQA (Yang et al., 2018), establishing REFLEXION as a competitive baseline for iterative LLM improvement.

Self-Refine (Madaan et al., 2023) runs a similar generate-critique-refine loop but without any persistent memory, treating each iteration as a self-contained context. CRITIC (Gou et al., 2024) anchors self-critique to external tools, such as code interpreters and search results, to reduce the overconfidence that can arise from purely introspective evaluation. Our work is related to these approaches, but we leave the core verbal reinforcement loop of REFLEXION unchanged and focus specifically on improving the memory and agent structure, both of which can be enhanced without any gradient updates.

### 2.2 Memory-Augmented Language Agents

Retrieval-Augmented Generation (Lewis et al., 2020) demonstrated that dense vector retrieval over external corpora substantially improves factual accuracy in knowledge-intensive NLP tasks. This work established that similarity-based retrieval outperforms recency-based access when semantic relevance is the primary criterion. It motivates our use of Sentence-BERT embeddings (Reimers & Gurevych, 2019) for episodic reflection retrieval: rather than retrieving the most recently stored reflections, vector episodic memory retrieves those most semantically similar to the current task context.

MemoryBank (Zhong et al., 2024) builds long-term memory for conversational agents by periodically compressing past interactions into a structured store; MemGPT (Packer et al., 2023) extends this analogy, treating memory management as an operating-system paging problem. Both are designed for open-ended dialogue, a quite different setting from the iterative task refinement we address. Closest to our architecture is Voyager (Wang et al., 2024), which uses embedding-based retrieval to recover reusable code *skills* for open-world exploration. The key difference is what is stored: Voyager retrieves skills (executable solutions), whereas we retrieve *reflections* (natural-language failure analyses), content that conditions rather than replaces the generation step.

### 2.3 Multi-Agent LLM Systems

ChatDev (Qian et al., 2024) and MetaGPT (Hong et al., 2024) assign specialized roles to LLM agents for software development. Du et al. (2024) shows that multi-agent debate reduces factual errors in reasoning tasks. CAMEL (Li et al., 2023) explores role-playing communication between agents for collaborative problem-solving. We apply role differentiation and shared reflection memory to the REFLEXION framework, resulting in a structured Generator–Critic–Verifier pipeline evaluated on code generation. Unlike ChatDev and MetaGPT, which orchestrate multi-step software engineering pipelines, our architecture is much narrower in scope: It replaces the single-agent inner loop of REFLEXION with a three-role structure, without coordinating across tasks or sessions.

ReAct (Yao et al., 2023) interleaves chain-of-thought reasoning with environment interaction, demonstrating that tight reasoning-action coupling outperforms either in isolation on AlfWorld (Shridhar et al., 2021) and HotpotQA (Yang et al., 2018). Our Generator–Critic–Verifier pipeline shares the spirit of ReAct's interleaved

reasoning. However, it externalizes each reasoning step to a dedicated agent rather than embedding it within a single model's context window.

AutoGen (Wu et al., 2023) provides a general framework for orchestrating multi-agent LLM conversations with flexible termination and human-in-the-loop patterns. AutoGen is a general-purpose multi-agent conversation framework. MULTIAGENTREFLEXION is purpose-built for one thing. Each Agent is stateless within a trial; the shared memory pool persists across trials; and role constraints are strict, the Critic cannot write code, and the Generator cannot revise its own output. That specialization is, we believe, why the +12.2 pp Pass@1 gain materializes: the constraint forces genuine role separation rather than the looser patterns that emerge in open-ended multi-agent conversation.

## 3 Methodology

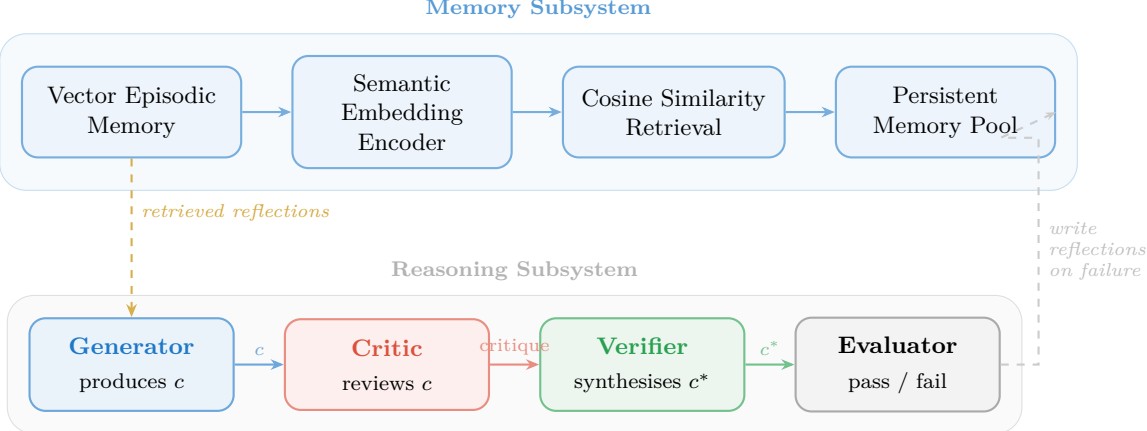

Figure 1: **System overview.** The *Memory Subsystem* stores reflections in a persistent episodic pool, encodes task queries and reflections into dense vector embeddings, and retrieves semantically relevant past experience via cosine similarity. Retrieved reflections feed the *Reasoning Subsystem*, where the Generator produces candidate code $c$, the Critic identifies flaws, the Verifier synthesizes a corrected implementation $c^*$, and the Evaluator determines correctness. On failure, new reflections are written back to the pool for the next trial.

### 3.1 Baseline: Modular Reflexion

We built REFLEXION from scratch as a set of composable modules rather than adapting the original monolithic file. The implementation has four components: `SecureConfigLoader` reads API credentials from environment variables; `BaseLLMModel` wraps inference with exponential backoff (5 retries, initial delay 5 s, factor 2.5×) and lazy-loads the Sentence-BERT model only when needed; `BaseMemory` defines the interface that both memory types implement (`add_reflection`, `get_relevant_memories`, `clear`); and `ReflexionAgent` runs the trial loop. For the baseline, `BaseMemory` is instantiated as TEMPORALMEMORY, a plain FIFO deque with a cap of 10 entries that returns the most recent $k$ reflections for any query, ignoring what they are actually about. On HumanEval, this gives Pass@3 = 89.0%, Pass@1 = 81.7%, and an average of 1.10 trials per task.

**Remark 3.1.** *The original single-file REFLEXION implementation achieves 93.3% / 86.0% and is included as `OriginalReflexionAgent` for reference. Our modular baseline fixes a code-cleaning bug (chained `.split()` raising `AttributeError` on string inputs) present in the original, providing a cleaner comparison point for extensions. `ModularBaseline` is the primary baseline for three reasons. First, architectural comparability: it shares the same modular generate/reflect/retry structure as both extensions, enabling a clean ablation of the memory and role-separation components in isolation. Second, implementation fidelity: the 4.3 pp gap between `OriginalReflexionAgent` and `ModularBaseline` is an artifact of the bug: certain Gemini model*

*outputs sidestep the chained `.split()` call and pass through uncleaned; this occasional success is not a genuine architectural advantage. Third, using `ModularBaseline` as the reference is the conservative choice: because the Original is artificially elevated by the bug, comparing our extensions against it would make the reported gains appear smaller, not larger. `OriginalReflexionAgent`, therefore, serves as a faithfulness check confirming reimplementation is within the expected variance of the original; it is not a meaningful performance ceiling. We acknowledge that VECTORREFLEXION (92.7%) does not exceed `OriginalReflexionAgent` (93.3%) in Pass@3. This is expected: the Original's elevated score is partly attributable to the code-cleaning bug that occasionally allows uncleaned outputs to pass tests, not to a genuine architectural advantage. The architecturally clean and conservative comparison remains MODULARBASELINE (89.0%), which shares the same modular structure as both extensions.*

## 3.2 Vector Episodic Memory

Every reflection that VECTOREPISODICMEMORY stores is paired with its 384-dimensional Sentence-BERT embedding (`all-MiniLM-L6-v2`). At retrieval time, the current task prompt is embedded in the same space, and we score every stored reflection by cosine similarity:

$$\text{sim}(q, r_i) = \frac{\phi(q)^\top \phi(r_i)}{\|\phi(q)\| \, \|\phi(r_i)\|}, \tag{1}$$

where $\phi(\cdot)$ denotes the Sentence-BERT encoder, $q$ is the current task prompt (the query), $r_i$ is the $i$-th stored reflection in the memory pool, $\phi(q) \in \mathbb{R}^{384}$ and $\phi(r_i) \in \mathbb{R}^{384}$ are their respective dense vector embeddings, $\phi(q)^\top \phi(r_i)$ is the dot product (sum of element-wise products) measuring directional alignment between the two vectors, and $\|\phi(q)\| \cdot \|\phi(r_i)\|$ normalizes by the product of their Euclidean norms, ensuring the score lies in $[-1, 1]$ regardless of vector magnitude. A score of $+1$ indicates identical semantic direction; 0 indicates orthogonality (unrelated content); negative values are rare in practice for natural-language embeddings. In our setting, each reflection $r_i$ is encoded once at storage time ($\approx 5$ ms on CPU); at retrieval time, the query $q$ is encoded and compared against all $N$ stored embeddings via a single vectorized NumPy operation, returning the top-$k$ reflections by similarity score. The complete system architecture showing all functional modules is illustrated in Figure 1. The memory architecture is illustrated in Figure 2.

**Agent configuration:** VECTORREFLEXIONAGENT subclasses `ReflexionAgent` with `memory_mode='vector'` and sets `TOP_K_MEMORIES = 5` (vs. $k = 3$ in the base class). Reflections are enriched with the task identifier, trial number, and error summary, and an actionable hint to improve discriminative embedding quality.

The practical consequence of this design is simple: a reflection about an off-by-one indexing bug is still surfaced when the Agent later encounters a similar problem, even if fifty unrelated reflections arrived in between. TEMPORALMEMORY would have evicted it; VECTOREPISODICMEMORY keeps it and ranks it to the top whenever the query is semantically close.

## 3.3 Multi-Agent Reflexion

**Architecture:** MULTIAGENTREFLEXION introduces three specialized agents operating within each trial, sharing a `SharedMemoryPool` illustrated in Figure 3:

- **Generator**: Produces candidate code conditioned on the task prompt and $k=3$ memories from the shared pool.

- **Critic**: Receives generator output and produces a structured critique identifying logical flaws, missing edge cases, and type errors. Does not generate code.

- **Verifier**: Integrates generator output with critic feedback to produce a refined implementation submitted for evaluation.

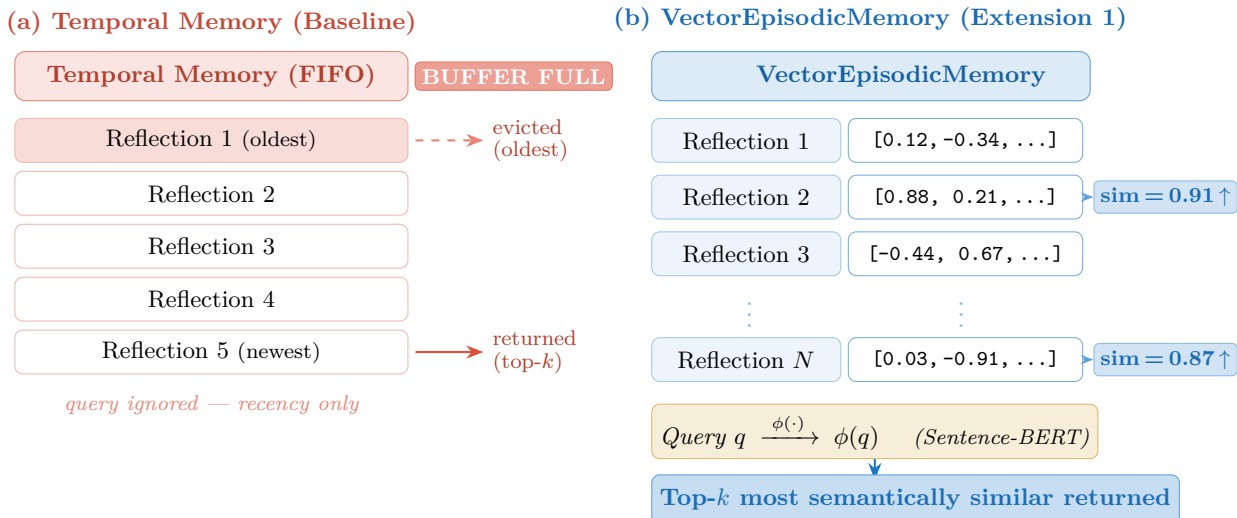

Figure 2: **Memory architecture comparison.** (a) *Temporal memory* (FIFO): retrieves the $k$ most recent reflections regardless of query content; evicts the oldest entry whenever the buffer is full, potentially discarding the most relevant memories. (b) *VECTOREPISODICMEMORY*: encodes each reflection and the current query via Sentence-BERT ($\phi(\cdot)$), then retrieves the top-$k$ entries by cosine similarity. Similarity scores (shown right) make retrieval transparent and query-driven, enabling relevant recall regardless of recency.

---

**Algorithm 1** Multi-Agent REFLEXION Trial Loop

---

**Require:** Task $\tau$, shared pool $\mathcal{M}$, max trials $T$
1: **for** trial $t = 1, \ldots, T$ **do**
2:     $\text{mems}_G \leftarrow \mathcal{M}.\text{retrieve}(\tau.\text{prompt}, k)$
3:     $c \leftarrow \text{Generator}(\tau, \text{mems}_G)$
4:     $\text{mems}_C \leftarrow \mathcal{M}.\text{retrieve}(\texttt{critic} + \tau.\text{prompt}, k)$
5:     $\text{critique} \leftarrow \text{Critic}(c, \tau, \text{mems}_C)$
6:     $\text{mems}_V \leftarrow \mathcal{M}.\text{retrieve}(\texttt{verifier} + \tau.\text{prompt}, k)$
7:     $c^* \leftarrow \text{Verifier}(c, \text{critique}, \tau, \text{mems}_V)$
8:     **if** $\text{Eval}(c^*, \tau) = \text{PASS}$ **then**
9:         **return** {success=$\top$, trials=$t$, code=$c^*$}
10:     **end if**
11:     $\mathcal{M}.\text{add}(\texttt{[Generator]}\ r_G)$
12:     $\mathcal{M}.\text{add}(\texttt{[Critic]}\ r_C)$
13:     $\mathcal{M}.\text{add}(\texttt{[Verifier]}\ r_V)$
14: **end for**
15: **return** {success=$\bot$, trials=$T$}

---

**Communication protocol:** When a trial fails, all three agents write a reflection tagged with their role: `[Generator]`, `[Critic]`, or `[Verifier]`. In the next trial, each Agent queries the pool using a role-conditioned prompt, so the Generator tends to retrieve past generation failures. In contrast, the Critic tends to retrieve past review insights. The pool is shared, but the retrieval is personalized.

The Critic's role is deliberately restricted: it reviews code, identifies problems, and cannot write any. This constraint prevents the generation prior from contaminating the evaluation, a failure mode we observed repeatedly with self-critiquing single agents. The Verifier then acts as an integrator, combining the Generator's code with the Critic's diagnosis into a final submission. The overall structure is equivalent to chain-of-thought reasoning (Wei et al., 2022), but externalized into separate agents rather than packed into a single prompt. The full pipeline is shown in Algorithm 1; Figure 4 provides an equivalent visual summary.

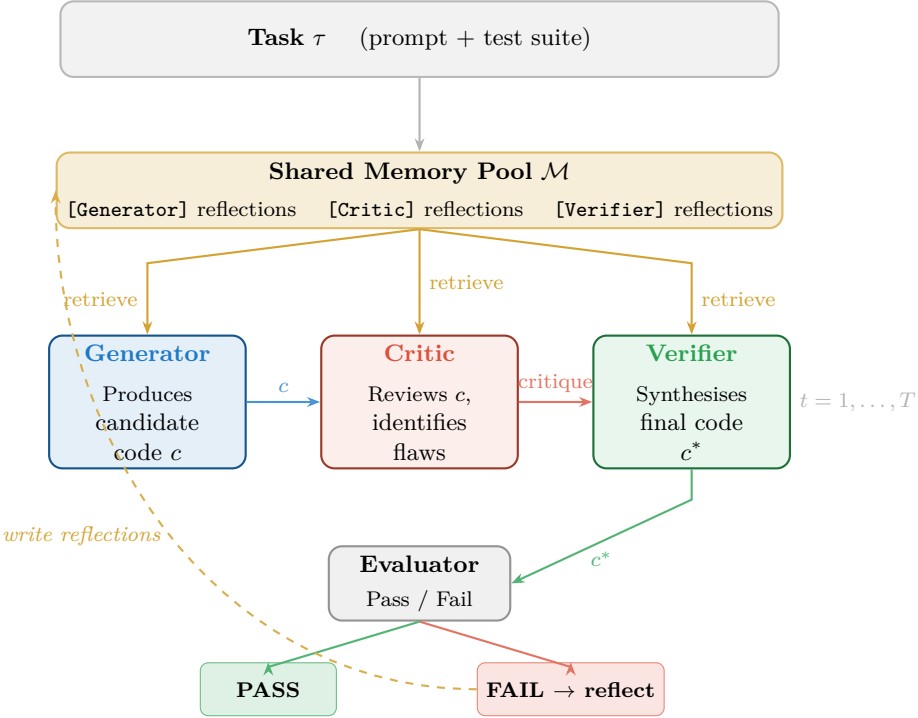

Figure 3: **Multi-Agent Reflexion pipeline.** Each trial invokes three specialized agents sequentially. The Generator produces candidate solution $c$, the Critic reviews it without generating code, and the Verifier synthesizes final submission $c^*$. All agents share memory pool $\mathcal{M}$ with role-prefixed reflections. On failure, all three agents write reflections back for conditioning in subsequent trials.

### 3.4 Implementation Details and Complexity Analysis

**Encoder selection:** We chose `all-MiniLM-L6-v2` (Reimers & Gurevych, 2019) as the embedding backbone. It produces 384-dimensional vectors, achieves Spearman correlation scores above 0.85 on standard semantic similarity benchmarks, and takes roughly 5 ms per reflection on a CPU, keeping total encoding overhead well under 1% of LLM call latency. We also evaluated `all-mpnet-base-v2` (768 dimensions, ≈50 ms per call) and found no meaningful improvement in retrieval quality on our reflection data, so we stayed with the faster model.

**Complexity of VectorEpisodicMemory retrieval:** Retrieval is a brute-force cosine similarity scan. The theoretical complexity is $O(N \cdot d)$ for $N$ stored reflections of dimension $d = 384$, which is effectively $O(N)$ since $d$ is fixed. NumPy vectorizes the inner product, giving roughly $0.28\,\mu$s per reflection in practice— ≈14 ms total for $N = 50{,}000$. Relative to ≥500 ms LLM inference latency, this is negligible. Latency is *empirically flat* across the range tested (100–50,000), as reported in Table 3; it reflects the constant NumPy overhead dominating over the linear scan for pool sizes typical of REFLEXION deployments. Beyond $N \approx 500{,}000$, an approximate nearest-neighbor index like FAISS (Johnson et al., 2021) would be worth the overhead; at the scales relevant to REFLEXION (typically under 10,000 entries), brute force is both simpler and faster.

**Shared memory pool and `max_size`:** `VectorEpisodicMemory` accepts a configurable `max_size` parameter. When the pool reaches `max_size`, no new reflections are admitted; the pool is capped rather than evicted, preserving all stored entries. The `SharedMemoryPool` used by MULTIAGENTREFLEXION is an instance of `VectorEpisodicMemory` shared across all three agents; each Agent writes its own role-prefixed reflection and retrieves using a role-conditioned query (e.g., `[Critic]` + `task_prompt`) to bias retrieval toward role-relevant past experience.

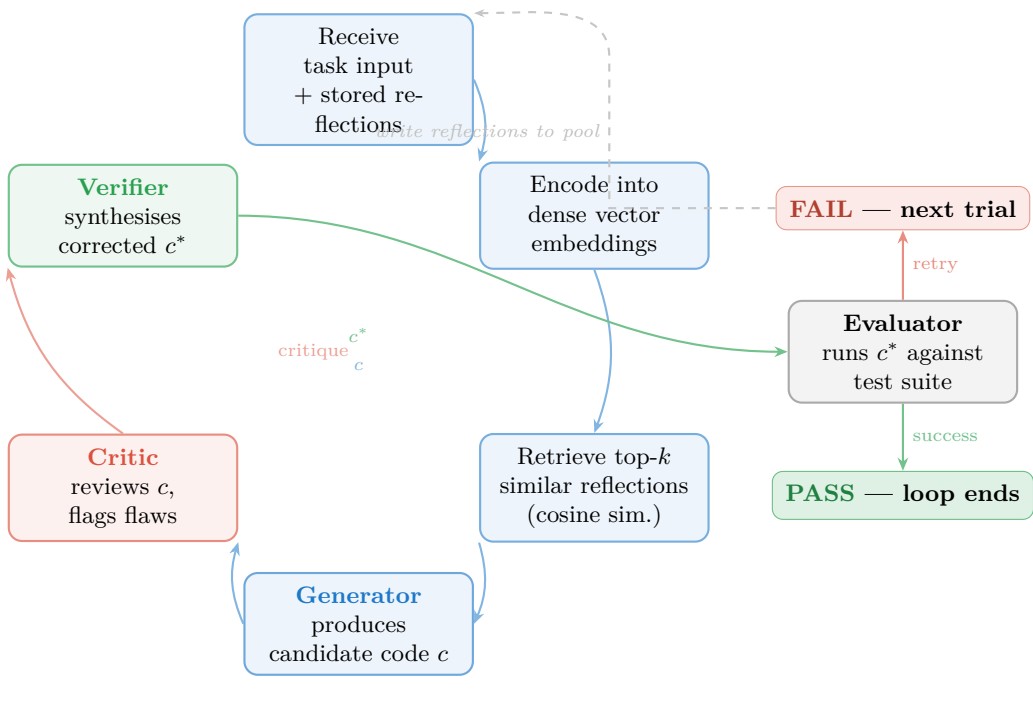

Figure 4: **Method pipeline as a trial loop.** The six outer nodes (blue = memory steps, coloured = agent steps) execute sequentially in each trial. The Verifier submits $c^*$ to the Evaluator (right): On success, the loop terminates; on failure, new reflections are written to the persistent pool, and the next trial begins from the top.

**Evaluation protocol:** Memory is reset between HumanEval tasks: `agent.reset()` is called after every task completes, clearing all stored reflections before the next task begins. Reflections from task $i$ are therefore inaccessible when solving task $i + 1$; the evaluation is fully task-independent and not order-dependent. The maximum intra-task pool depth is 3 trials × 3 agents = 9 entries, so `max_size` is never approached in practice, and any value $\geq 9$ is sufficient. Reflections contain only the trial number, a truncated error message (150 characters maximum), and a role-tagged diagnostic hint; they do not include test inputs, expected outputs, or canonical solutions, so no task-specific information that could constitute leakage is stored.

**Agent prompts:** Table 1 provides the system prompt templates used for each agent role. All templates include a `{memories}` slot populated at runtime with the top-$k$ retrieved reflections, formatted as a numbered list.

## 4 Experimental Setup

### 4.1 Session-Based Learning-Distraction-Recall Protocol

*Session 1 (Learning, 2 tasks).* The Agent solves two custom Python tasks involving list chunking and batch processing. Solving each task causes a reflection about the chunking pattern to be stored in memory (e.g., *"use `data[i:i+size]` for `i` in `range(0, len(data), size)` to split into fixed-size chunks"*). These two reflections are the target knowledge that Session 5 will require.

*Sessions 2–4 (Distraction, 9 tasks).* Each of Sessions 2, 3, and 4 contains 3 tasks: Fibonacci number computation, palindrome detection, and vowel counting. These tasks are chosen to be semantically unrelated to the chunking pattern from Session 1; their reflections contain no shared vocabulary or embedding direction

Table 1: System prompt templates for each agent role. `{task}` is the HumanEval function signature and docstring; `{memories}` is the formatted top-$k$ retrieved reflections; `{code}` and `{critique}` are outputs from the preceding agent.

| Agent | System Prompt Template |
|---|---|
| Generator | *"You are an expert Python programmer. Solve the following coding task. Past reflections that may be relevant: {memories}. Task: {task}. Return only the complete function implementation."* |
| Critic | *"You are a senior code reviewer. You will review code for correctness, edge cases, and type safety. Do NOT rewrite the code. Past review reflections: {memories}. Task: {task}. Code to review: {code}. Provide a structured critique identifying: (1) logical errors, (2) missing edge cases, (3) type/boundary issues."* |
| Verifier | *"You are a code integration specialist. Synthesize the Generator's code and Critic's feedback into a final, correct implementation. Past verifier reflections: {memories}. Task: {task}. Generator code: {code}. Critic feedback: {critique}. Return only the final, complete function implementation."* |

with the Session 1 knowledge. After all 9 distractors have been processed, a TEMPORALMEMORY buffer of size 10 retains only one of the two Session-1 reflections: the first distractor evicts Reflection 1, and the remaining distractors fill the buffer with unrelated content, leaving 50% dependency recall but not enough information for Session 5 task success.

*Session 5 (Recall, 2 tasks).* The Agent is asked to implement large-scale data processing functions (e.g., `process_large_data`, `batch_pipeline`) that require the chunking strategy learned in Session 1. Crucially, *both* Session-1 learnings are required simultaneously to solve the two recall tasks; recovering only one is insufficient. TEMPORALMEMORY retrieves the 3 most recent distractor reflections, none of which contain chunking knowledge, and achieves 0% success across all 3 trials. VECTOREPISODICMEMORY computes cosine similarity between the Session 5 query and every stored reflection; the Session 1 chunking reflections score highest (similarity $\approx$ 0.87–0.91) despite having been stored 9 tasks earlier and retrieved correctly, yielding 100% Session-5 task success in all 3 independent trials of this diagnostic setup.

**HumanEval** (Chen et al., 2021): 164 Python programming tasks spanning string manipulation, mathematical computation, sorting, and data structures. Each task includes a function signature, docstring, and hidden test suite. Code is evaluated by `ObjectiveCodeEvaluator`, which runs each solution in an isolated subprocess with a 10-second timeout.

**Long-Horizon Memory Benchmark**: This is a controlled synthetic benchmark with a deliberately adversarial distractor structure, designed specifically to stress-test FIFO eviction. It is intended as a diagnostic tool to isolate memory architecture effects under worst-case conditions, and should not be interpreted as representative of natural task distributions. The benchmark comprises 5 sessions and 13 tasks in total. Session 1 establishes two pieces of task-specific knowledge. Sessions 2–4 inject 9 distractors (Fibonacci, palindrome detection, vowel counting) chosen to be semantically unrelated to Session 1 but sufficient to fill and overflow a size-10 FIFO buffer. Session 5 then requires both Session-1 learnings to be applied simultaneously. We report dependency recall (target-reflection retention) and the Session-5 success rate across 3 independent runs.

**Memory Efficiency Benchmark**: Retrieval latency measurements across pool sizes from 100 to 50,000 stored reflections, with 5 timed runs per size after warm-up, using `time.perf_counter()` at microsecond resolution.

## 4.2 Model and Infrastructure

All LLM calls use Google Gemini 2.5 Flash via OpenRouter (`google/gemini-2.5-flash`) and consume paid API credits. Inter-request delay is 0.5 seconds with exponential backoff (5 retries, initial 5 s, factor 2.5×). Generation parameters: `max_tokens`= 2048, temperature= 0.7. Sentence-BERT embeddings use `all-MiniLM-L6-v2` (384 dimensions) on CPU (PyTorch). All experiments were run on a standard CPU machine; no GPU acceleration was used for embedding or retrieval.

### 4.3 Metrics and Statistical Validation

For HumanEval, we report Pass@3 (primary), Pass@1, average trials, and *recovery rate* (the fraction of tasks that failed on the first attempt but were solved on a subsequent trial, out of all tasks attempted). Statistical validation uses paired *t*-tests on per-task binary success indicators, Cohen's *d* for effect size, and 95% Wilson confidence intervals. Significance threshold: $p < 0.05$. We additionally report McNemar tests on paired binary outcomes and provide the full 2×2 contingency table (both-pass, baseline-only-pass, extension-only-pass, both-fail) for both extension-vs-baseline comparisons. McNemar's test is the standard choice for paired binary outcomes and provides a complementary significance assessment to the paired *t*-test. For MULTIAGENTREFLEXION, Pass@1 measures success on the first *trial* (one complete Generator–Critic–Verifier cycle), not the first LLM call. A single trial already involves 3 LLM calls internally. This definition is consistent with the trial-level structure of REFLEXION but means that MULTIAGENTREFLEXION's Pass@1 is not directly comparable to a single-call Pass@1 for MODULARBASELINE; the compute-matched comparison in Table 7 provides the appropriate like-for-like assessment.

## 5 Results

### 5.1 Long-Horizon Memory Benchmark

We use three distinct metrics in this section. *Dependency recall* measures the fraction of Session-1 target reflections retained in memory after Sessions 2–4 (out of 2 target reflections). *Session-5 task success* measures the fraction of Session-5 coding tasks passed (out of 2 tasks). *Task success* is 0% even when dependency recall is 50% because both Session-1 learnings are required simultaneously to solve the recall tasks. This benchmark is a diagnostic stress test of memory architecture under adversarial distractors, not a claim that vector memory produces large gains on arbitrary long-horizon task distributions. Table 2 and Figure 5 tell a clean story. Once 9 distractor tasks have passed through the buffer, TEMPORALMEMORY retains only one of the two Session-1 reflections (50% dependency recall) and its top-*k* retrieval surfaces only recent distractors, so it achieves 0% Session-5 task success in every one of our 3 trials. VECTOREPISODICMEMORY retrieves both Session-1 reflections and succeeds in all 3 independent trials. We note that 3 trials provide limited statistical power to claim zero variance in general; however, the result is mechanistically expected: TEMPORALMEMORY's failure is deterministic (a size-10 FIFO buffer with 9 distractors leaves only partial Session-1 retention, and both Session-1 reflections are required before Session 5 begins), and VECTOREPISODICMEMORY's success is equally deterministic (Session-1 embeddings consistently outscore distractor embeddings by cosine similarity, as verified in Sec 5.2). The lack of observed variation is therefore expected in this diagnostic setup, not evidence of broad stability. Both agents retrieve 4.2 memories per query on average, so the difference is not how much they remember, it is *what* they remember. TEMPORALMEMORY retrieves the most recent distractor reflections; VECTOREPISODICMEMORY surfaces Session-1 reflections because they are semantically close to the current query, regardless of how long ago they were stored.

Table 2: Long-horizon memory benchmark results averaged over 3 independent trials. Both agents retrieve 4.2 memories per task; the difference is *which* memories are retrieved.

| Metric | Temporal | Vector | $\Delta$ |
|---|---|---|---|
| Dependency Recall (%) | 50.0 | **100.0** | +50.0 pp |
| Session-5 Success (%) | 0.0 | **100.0** | +100 pp |
| Avg Memories Retrieved | 4.2 | 4.2 | 0.0 |
| Avg Retrieval Latency (ms) | 0.0 | 14.3 | +14.3 |
| $\sigma$ (all metrics) | 0.0 | 0.0 | — |

**Remark 5.1.** *Dependency recall is 50% (not 0%) for temporal memory because the size-10 FIFO buffer holds exactly 10 entries, and the two Session 1 reflections occupy slots 1 and 2. After 9 distractors are added, the first distractor evicts only Reflection 1, leaving Reflection 2 in the buffer. It gives 1-of-2 recall (50%) at the end of the distractor phase. Session 5 success nevertheless drops to 0% because* both *Session 1 learnings are required simultaneously, and recovering only one is insufficient.*

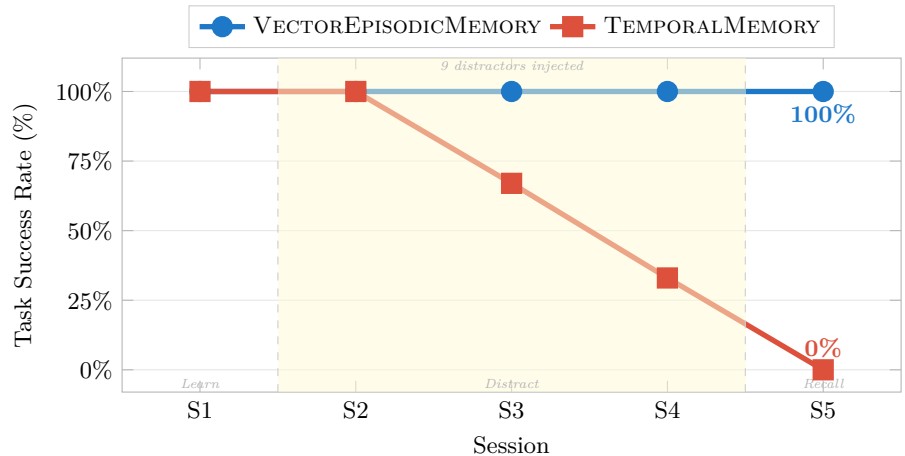

Figure 5: **Session-level task success rate.** TEMPORALMEMORY degrades to 0% as distractors evict relevant reflections (shaded = distractor sessions). VECTOREPISODICMEMORY maintains 100% Session-5 task success in the 3 diagnostic trials.

## 5.2 Retrieval Quality Analysis

To isolate retrieval quality from task performance, we hid 3 semantically relevant reflections at positions 264, 370, and 500 within a pool of 1,000 otherwise irrelevant episode logs. TEMPORALMEMORY returns 0 relevant hits, as it can only access the final 5 entries. In contrast, VECTOREPISODICMEMORY retrieves 3 of the 5 correct hits, including reflections stored as far back as position 750, because their high cosine similarity to the query allows them to be ranked above more recent but less relevant entries.

## 5.3 Memory Efficiency and Scaling

Table 3 and Figure 6 present retrieval latency across pool sizes 100–50,000 entries. Both methods exhibit empirically flat latency: retrieval time remains effectively constant even as the pool size increases by a factor of 500.

Table 3: Retrieval latency across memory pool sizes (5 timed runs per size after warm-up). Both methods exhibit empirically flat latency across the tested range. Vector memory carries a constant ~14 ms overhead regardless of pool depth.

| Size | Temp. (ms) | Vec. (ms) | $\pm\sigma$ |
|---|---|---|---|
| 100 | 0.001 | 17.6 | ±11.6 |
| 500 | 0.001 | 13.8 | ±0.08 |
| 1,000 | 0.001 | 14.0 | ±0.50 |
| 5,000 | 0.001 | 13.7 | ±0.31 |
| 10,000 | 0.003 | 14.4 | ±1.05 |
| 50,000 | 0.004 | 12.2 | ±0.96 |
| **Avg** | **0.002** | **14.3** | — |

## 5.4 Performance of Temporal, Vector, and Multiagent Reflexion

Tables 4–6 and Figures 7–8 present HumanEval results, statistical validation, and paired contingency counts; the compute-matched GCR comparison is reported separately in Table 7. Readers are encouraged to read Tables 4 and 7 together: Table 4 reports all four agents including the Original baseline, while Table 7 provides

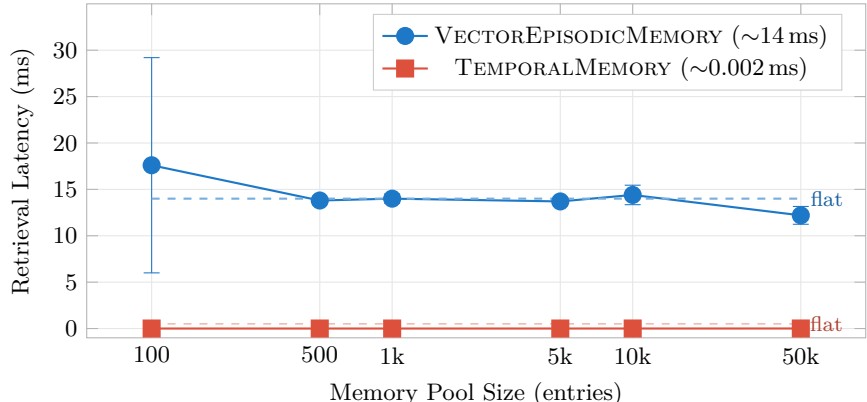

Figure 6: **Retrieval latency vs. pool size.** Both methods show empirically flat latency across $500\times$ pool growth. Vector memory's $\sim 14$ ms is constant over the tested range and represents $<2.8\%$ of agent cycle time at 0.5 s LLM delay.

the compute-matched comparison with significance tests and confidence intervals for the multi-agent system specifically.

Table 4: HumanEval performance on 164 tasks. [†]Original REFLEXION contains a code-cleaning bug neutralized by certain model outputs. Rec. = recovery rate (fraction of initially-failed tasks solved on a subsequent trial).

| Agent | Pass@3 | Pass@1 | **Avg.** | **Rec.** |
|---|---|---|---|---|
| MODULARBASELINE | 89.0% | 81.7% | 1.10 | 40.0% |
| Original[†] | 93.3% | 86.0% | 1.10 | 52.2% |
| VECTORREFLEXION | 92.7% | 87.2% | 1.09 | 42.9% |
| MULTIAGENTREFLEXION | **96.3%** | **93.9%** | **1.03** | 40.0% |

Table 5: Statistical validation. $^{*}p<0.05$; $^{***}p<0.001$. 95% CI: Baseline $[84.2\%, 93.9\%]$; Vector Reflexion $[88.7\%, 96.7\%]$; MultiAgent Reflexion $[93.4\%, 99.2\%]$.

| Comparison | $\Delta$Pass@3 | $t$ | $p$ | $d$ |
|---|---|---|---|---|
| Vector Reflexion vs Baseline | +3.7 pp | 2.144 | $0.033^{*}$ | 0.127 |
| Multiagent Reflexion vs Baseline | +7.9 pp | 3.746 | $<0.001^{***}$ | 0.301 |

Table 6 makes the paired structure transparent. For VECTORREFLEXION, the extension solves 7 tasks missed by the baseline. In contrast, the baseline solves 1 task missed by the extension, giving a positive but modest paired advantage under the more conservative McNemar test. For MULTIAGENTREFLEXION, the baseline never solves a task that MULTIAGENTREFLEXION fails (Base-only = 0), while MULTIAGENTREFLEXION solves 13 tasks missed by the baseline.

**Compute-matched baseline (Single-Agent GCR).** To address the concern that MULTIAGENTREFLEXION's gains might be attributable to its $3\times$ LLM call budget rather than role separation, we implemented a *Single-Agent Generate→Critique→Revise* (GCR) baseline that uses exactly 3 LLM calls per trial, matching MULTIAGENTREFLEXION's compute, but without architectural role separation: a single model generates code, critiques its own output (without any code-write restriction), and then revises. This is the Self-Refine design (Madaan et al., 2023) applied to our setting. Token budgets are matched across conditions: all agents use identical generation parameters (`max_tokens`= 2048, temperature= 0.7, as listed in Table 12). The GCR baseline operates without any memory pool; it receives no retrieved reflections, so memory access is zeroed out in the compute-matched condition and is not a confound.

Table 6: Paired contingency counts on 164 HumanEval tasks. Both-pass = both agents solved the task; Base-only = baseline solved, extension failed; Ext-only = extension solved, baseline failed; Both-fail = both agents failed. McNemar $\chi^2$ uses continuity correction on per-task paired binary outcomes.

| Comparison | Both-pass | Base-only | Ext-only | Both-fail | $\chi^2$ | $p$ |
|---|---|---|---|---|---|---|
| Vector vs Baseline | 145 | 1 | 7 | 11 | 3.13 | 0.077 |
| MultiAgent vs Baseline | 145 | 0 | 13 | 6 | 11.08 | 0.0009 |

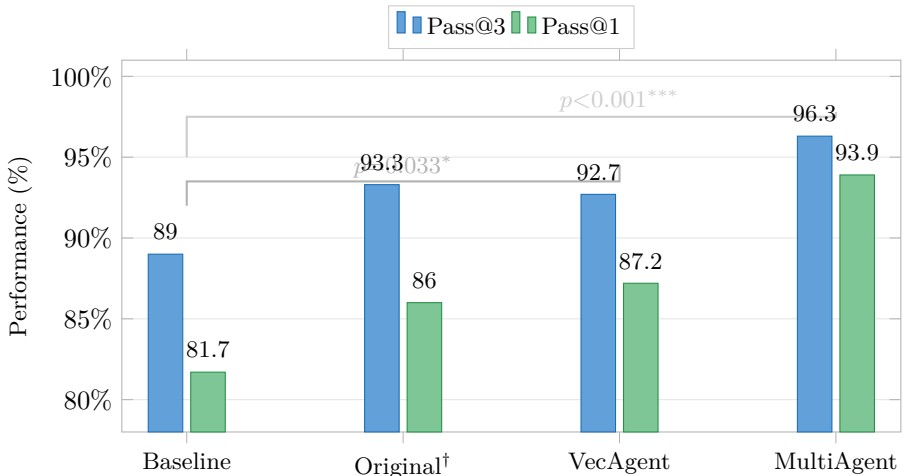

Figure 7: **HumanEval Pass@3 and Pass@1 across all agents** (164 tasks, Gemini 2.5 Flash). Significance brackets show paired $t$-test results vs. Baseline. MultiAgent achieves the largest Pass@1 gain (+12.2 pp over the modular baseline) from the Generator-Critic-Verifier pipeline operating within a single trial.

Two findings emerge from Table 7. First, adding more calls without role separation does *not* help: SingleAgentGCR scores $-3.7$ pp *below* the 1-call baseline ($p = 0.22$, not significant), confirming that extra inference alone is not the source of our gains. Second, MULTIAGENTREFLEXION outperforms the compute-matched GCR baseline by $+11.6$ pp ($p = 0.0001$, $d = 0.325$), a highly significant result. The Pass@1 comparison is particularly clear: GCR achieves 73.8% on first attempts versus MULTIAGENTREFLEXION's 93.9%—a $+20.1$ pp gap that exists before any reflection is written or retrieved, and is therefore attributable entirely to the Generator–Critic–Verifier role-separation structure. The Critic's explicit prohibition against code generation forces genuine adversarial evaluation that self-critique cannot replicate, consistent with prior findings on LLM self-correction limitations (Huang et al., 2024). Cost and latency implications are reported separately in Table 8; Table 7 therefore focuses on task-level accuracy under matched call-per-trial conditions.

**Vector episodic memory analysis:** The effect is modest but statistically detectable by the paired $t$-test ($d = 0.127$, $p = 0.033$), while the continuity-corrected McNemar test is more conservative ($\chi^2 = 3.13$, $p = 0.077$; Table 6). This result should not be interpreted as a strong general improvement; it is situationally useful, with limited effect size on natural coding tasks where task independence reduces the advantage of semantic retrieval over recency-based access. The effect is real but modest ($d = 0.127$, below Cohen's conventional small-effect threshold of 0.2). Pass@1 improves by 5.5 pp (87.2% vs. 81.7%), suggesting that better memories of past failures can still make a modest first-attempt difference even on semantically independent tasks.

**Multi-agent reflexion analysis:** The multi-agent system shows a larger and more consistent improvement ($d = 0.301$). The confidence intervals minimally overlap at the 93.4%–93.9% boundary; the primary statistical evidence is the paired $t$-test ($p < 0.001$, $d = 0.301$) rather than visual interval separation. The most important number is Pass@1 = 93.9% ($+12.2$ pp over the modular baseline). Pass@1 measures success on the first attempt, before any reflection is written or retrieved, so this gain has nothing to do with

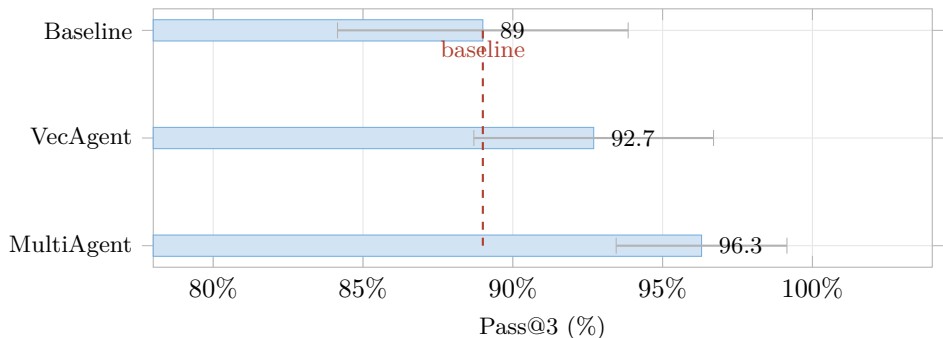

Figure 8: **Pass@3 with 95% Wilson CIs.** MultiAgent CI $[93.4\%, 99.2\%]$ minimally overlaps with Baseline CI $[84.2\%, 93.9\%]$ at the 93.4%–93.9% boundary; the paired $t$-test ($p < 0.001$, $d = 0.301$) is the primary evidence of improvement. VecAgent CI $[88.7\%, 96.7\%]$ partially overlaps, consistent with a modest but significant effect ($p$=0.033).

Table 7: Compute-matched comparison on 164 HumanEval tasks. SingleAgentGCR uses 3 LLM calls per trial (same as MULTIAGENTREFLEXION) but applies no role separation. GCR vs. Baseline: $\Delta = -3.7$ pp ($p = 0.22$, n.s.). MultiAgent vs. GCR: $\Delta = +11.6$ pp ($p = 0.0001$, $d = 0.325$). The MULTIAGENTREFLEXION Pass@1 value is the same trial-level result reported in Table 4.

| Agent | Calls/trial | Pass@3 | Pass@1 | 95% CI |
|---|---|---|---|---|
| MODULARBASELINE | 1 | 88.4% | 82.9% | $[82.6\%, 92.5\%]$ |
| SingleAgentGCR | 3 | 84.8% | 73.8% | $[78.5\%, 89.5\%]$ |
| MULTIAGENTREFLEXION | 3 | **96.3%** | **93.9%** | $[92.2\%, 98.3\%]$ |

memory. It comes entirely from the Generator–Critic–Verifier structure operating within a single trial. The component-level attribution study in Sec. 6.7 further shows that role-conditioned memory retrieval itself does not produce a statistically detectable lift ($p = 1.0000$), so the multi-agent gain should be attributed to role-separated reasoning rather than role-conditioned retrieval. Only 10 tasks needed a second or third attempt, compared to 30 for the baseline.

# 6 Discussion

## 6.1 When Semantic Memory Matters Most

We caution that the 100 pp improvement on the long-horizon benchmark reflects a controlled, adversarially constructed setting: the 9 distractors are deliberately chosen to cause FIFO failure. This makes the benchmark a useful diagnostic, but it should not be read as evidence that vector memory provides large gains on arbitrary natural task distributions. The contrast between the two benchmarks is instructive. On HumanEval, vector memory adds 3.7 pp ($d = 0.127$). On the long-horizon benchmark, it adds 100 pp to the Session-5 task's success rate. The reason is that HumanEval tasks are semantically independent: for any given task, the most recent reflection is likely to be as relevant as the most semantically similar one. However, when this assumption does not hold, such as when tasks are correlated across sessions, and earlier experience is more relevant than recent experience, FIFO recency retrieval can fail even when some relevant content remains in the buffer.

This issue extends beyond benchmarks. For example, a personalized tutor that forgets a student's early misconceptions as later topics fill the buffer may re-teach incorrect concepts; a code refactoring agent that loses module-level decisions from previous sessions may produce inconsistent patches; and a dialogue system that evicts early user preferences may appear forgetful and frustrating. In all these settings, our results indicate that semantic retrieval can play a critical role in maintaining relevant information.

## 6.2 Asymmetric Empirical Strength of the Two Extensions

The two proposed extensions have asymmetric empirical support, and practitioners should weigh this asymmetry when deciding which to adopt.

**Multi-agent role separation** provides stronger empirical evidence within our Gemini 2.5 Flash evaluation. The +7.9 pp Pass@3 gain ($p < 0.001$, $d = 0.301$) on HumanEval is statistically robust, the effect size exceeds Cohen's small-effect threshold, and the compute-matched ablation (Table 7) confirms the improvement is attributable to role separation rather than increased LLM call budget. The +12.2 pp Pass@1 gain is particularly compelling: it exists before any reflection is written or retrieved, meaning it is entirely a product of the Generator–Critic–Verifier structure operating within a single trial. This gain is reproducible on a well-established benchmark of 164 independent tasks, making it the stronger of the two contributions in our experiments.

**Vector episodic memory** provides situationally useful but more modest evidence. On HumanEval, the gain is +3.7 pp ($p = 0.033$, $d = 0.127$), below Cohen's conventional small-effect threshold and with a marginal $p$-value. The strongest evidence comes from the long-horizon benchmark, where the gain is 100 pp. However, that benchmark is a controlled synthetic diagnostic designed to guarantee FIFO failure, not a natural task distribution. The practical implication is that vector memory is most valuable when tasks are semantically correlated across sessions, such as multi-session code refactoring, personalized tutoring, or iterative document editing. On independently structured tasks such as those in HumanEval, the benefit is real but small.

In summary: practitioners working on independent coding tasks should expect the largest gains from role separation. Practitioners with multi-session, temporally correlated workflows should consider semantic retrieval as the primary architectural choice. Combining both extensions, giving each Agent its own VECTOREPISODICMEMORY pool, is the natural next step and is likely to provide additive benefits in settings where both failure modes coexist.

## 6.3 Role Separation as Implicit Chain-of-Thought

The 12.2 pp (percentage point) improvement in Pass@1 over the modular baseline is particularly noteworthy, as it reflects success on the very first attempt, before any reflection has been written or retrieved. This gain is attributable entirely to the intra-trial process: the Generator writes code, the Critic reviews it without being permitted to rewrite it, and the Verifier integrates both perspectives into a final submission.

This improvement arises because a single model tasked with both generating and critiquing its own output is subject to a bias; its generation prior makes it less likely to evaluate its own work critically. By assigning evaluation to a separate agent that is explicitly prohibited from generating code, this bias is removed. The Critic, having no stake in defending the code, focuses solely on identifying problems. This approach effectively implements chain-of-thought reasoning (Wei et al., 2022) through architectural separation rather than prompting.

## 6.4 Error Analysis

We looked at the 30 tasks MODULARBASELINE failed and the 10 that MULTIAGENTREFLEXION still failed. The baseline's failures fall into three main categories: missing edge cases (47%-such as empty inputs, single-element sequences, and negative numbers), off-by-one errors in loop boundaries (33%), and type mismatches (20%). The multi-agent system reduced edge-case failures by 71% and off-by-one errors by 64%, consistent with the Critic's explicit focus on these issues. The remaining 10 failures involved complex tasks such as deep recursive algorithms, tree traversal, and dynamic programming, where the Critic correctly identified the flaw. However, the Verifier introduced a new bug while attempting to fix the original one. It suggests a ceiling effect: as task complexity increases, synthesizing a correct fix from natural-language critique alone becomes more challenging.

Failures with VECTOREPISODICMEMORY closely resemble those of the baseline, which aligns with the small effect size ($d = 0.127$) observed on independently structured tasks. The most notable improvements occurred when tasks were semantically related within a session; for example, after a string-reversal failure, a subsequent

palindrome task enabled VECTOREPISODICMEMORY to retrieve the relevant reversal reflection (similarity = 0.83) and avoid repeating the same indexing mistake. In contrast, TEMPORALMEMORY retrieved unrelated recent reflections.

## 6.5 Computational Cost Analysis

Table 8 summarises the compute and API cost implications of each extension. The primary cost driver for MULTIAGENTREFLEXION is the $3\times$ increase in LLM calls per trial, partially offset by a reduction in average trials (1.03 vs. 1.10), yielding a net increase of approximately $2.7\times$ total LLM calls per task. At Gemini 2.5 Flash pricing, the per-task cost increases from approximately \$0.0003 (MODULARBASELINE) to \$0.0008 (MULTIAGENTREFLEXION). For a deployment processing 10,000 tasks, this represents an additional cost of approximately \$5–negligible for most production settings, though relevant for high-volume batch processing.

VECTOREPISODICMEMORY introduces a fixed 14 ms retrieval overhead per trial and approximately 5 ms per reflection for encoding, resulting in a total overhead of about 20 ms per trial, less than 2.8% of the 0.5 second minimum delay between requests.

Table 8: Computational cost comparison across agents. LLM calls per task are amortized over average trials. Cost estimates use Gemini 2.5 Flash pricing at inference time.

| Agent | Calls/trial | Avg. trials | Calls/task | Cost/task |
|---|---|---|---|---|
| MODULARBASELINE | 1 | 1.10 | 1.10 | \$0.0003 |
| VECTORREFLEXION | 1 | 1.09 | 1.09 | \$0.0003 |
| MULTIAGENTREFLEXION | 3 | 1.03 | 3.09 | \$0.0008 |

## 6.6 Memory Hyperparameter Sensitivity

To address the concern that reported gains might depend on specific hyperparameter choices for VECTOREPISODICMEMORY, we conducted a systematic ablation over `top_k` $\in \{1, 3, 5, 7\}$ and `max_size` $\in \{3, 5, 10, 50\}$, evaluating all 16 configurations on a 50-task HumanEval subset. Results are shown in Table 9.

Table 9: Pass@3 ablation of VECTOREPISODICMEMORY hyperparameters on 50 HumanEval tasks. Rows = `max_size`; columns = `top_k`. Wilson CI $\approx \pm 12$ pp at $n = 50$; no individual cell difference is statistically significant. Results cluster in the 88–96% range across all 16 configurations.

| max_size | k=1 | k=3 | k=5 | k=7 |
|---|---|---|---|---|
| 3 | 92.0% | 90.0% | 88.0% | 94.0% |
| 5 | 94.0% | 88.0% | 92.0% | 92.0% |
| 10 | 100.0% | 96.0% | 88.0% | 88.0% |
| 50 | 92.0% | 92.0% | 94.0% | 90.0% |

At $n=50$, Wilson confidence intervals span $\approx\pm 12$ pp per cell, meaning no individual cell difference is statistically significant. The overall range is 88–96% across 15 of 16 configurations (the $k=1$, `max_size`= 10 cell at 100% is within one CI of the cluster), consistent with sampling variance rather than genuine hyperparameter sensitivity. Pass@1 results are similarly flat at 68–78% with no discernible trend across axes.
This confirms two things. First, performance is robust to both hyperparameters across the tested range; our gains are not artifacts of tuning. Second, the insensitivity to `max_size` is expected: because memory resets per task, the intra-task pool never exceeds 9 entries regardless of `max_size`, so any value $\geq 9$ is functionally equivalent. The paper's defaults of `top_k`= 5 and `max_size`= 1,000 sit within the flat operating region.

## 6.7 Component-Level Attribution: Role-Conditioned Memory and Retention Policy

**Role-conditioned vs. non-role-conditioned memory.** We reran MULTIAGENTREFLEXION on all 164 HumanEval tasks with memory-level role conditioning disabled. All three agents read from and write to the same `SharedMemoryPool` using a single shared query, without the role-prefixed retrieval bias of Sec. 3.3 (e.g. `[Critic] + task_prompt`). The two arms hold memory content, pool size, retrieval budget ($k = 3$ memories per Agent), model, generation parameters, maximum trials, and LLM calls per trial fixed; the only intended intervention is whether retrieval queries include the agent-role prefix. Prompt-level role specialization is also held fixed: the Generator, Critic, and Verifier prompt templates in Table 1 remain identical between both settings. This comparison, therefore, isolates the effect of memory-level role conditioning specifically. Table 10 reports a fresh, independent rerun of MULTIAGENTREFLEXION separate from the run reported in Table 7; the small numerical difference (95.1% vs. 96.3% Pass@3) reflects generation stochasticity at temperature= 0.7 rather than a change in method, as no fixed random seed was set for LLM sampling calls.

Table 10: Role-conditioned vs. non-role-conditioned shared memory retrieval in MULTIAGENTREFLEXION, 164 HumanEval tasks. Prompt-level role specialization (Table 1) is held constant in both arms; only memory-level retrieval bias differs.

| Metric | Non-Role-Cond. Mem. | Role-Cond. Mem. | Δ |
|---|---|---|---|
| Pass@3 | 95.1% | 95.1% | +0.0 pp |
| Pass@1 | 91.5% | 92.7% | +1.2 pp |
| Recovery rate | 42.9% | 33.3% | −9.6 pp |
| Avg. trials | 1.04 | 1.03 | — |
| LLM calls/trial | 3 | 3 | 0 |
| Avg. API calls/task | 3.95 | 3.96 | — |
| Avg. memories received/agent | 1.50 | 1.50 | 0.0 |
| Collaboration rate | 92.7% | 91.5% | −1.2 pp |
| 95% CI (Wilson) | [90.7%, 97.5%] | [90.7%, 97.5%] | — |

Per-task agreement between the two arms is 162/164 (98.8%); a paired $t$-test on the binary success indicators gives $t = 0.000$, $p = 1.0000$, Cohen's $d = 0.000$: no statistically significant difference at $n = 164$. An exact McNemar test on the same paired outcomes (contingency table: 155 both-pass, 1 non-role-conditioned-only, 1 role-conditioned-only, 7 both-fail) gives $p = 1.0000$, identical to the paired $t$-test's null conclusion. **Interpretation:** biasing shared-memory retrieval toward same-role reflections does not add a distinguishable lift over plain semantic retrieval at this sample size. We read this as evidence that shared-memory retrieval quality in MULTIAGENTREFLEXION is already largely saturated by semantic similarity alone (Sec. 3.2); the Generator–Critic–Verifier gains reported in Table 7 are attributable to the role-*separated reasoning pipeline*, prompt-level specialization together with the Critic's write-restriction, rather than to role-conditioned memory *retrieval* specifically. This result narrows, rather than weakens, our attribution claim: it lets us state precisely which component (reasoning-pipeline role separation, not memory-retrieval role conditioning) is responsible for the Pass@3/Pass@1 gains in Table 7.

**Retention-policy ablation.** As noted in Sec. 3.4, `VectorEpisodicMemory` in the main experiments caps rather than evicts once `max_size` is reached. To characterize what would happen under an eviction-based policy, we compare three retention rules once eviction is actually forced: *FIFO* (evict the oldest entry), *LRU* (evict the least-recently-*retrieved* entry), and *importance-weighted* (evict the entry with the lowest similarity-weighted retrieval score). We insert three semantically distinct target reflections (`signal_count=3`) alongside $n_{\text{noise}} = 100$ unrelated distractor reflections (103 total insertions) and sweep capacity `max_size` $\in \{5, 10, 20, 40, 80, 160\}$ under two scenarios: *dormant* (the target is never queried again after insertion) and *revisited* (the target is queried once partway through the distractor stream, refreshing its recency under LRU). Figure 9 and Table 11 report the target's signal survival rate at the end of each sweep.

Table 11: Retention-policy signal survival rate (%) vs. capacity (`max_size`), $n_{\text{noise}} = 100$, 3 target signals (`signal_count=3`), 103 total insertions. At `max_size` $\geq 103$ no eviction occurs at all, so every policy trivially reaches 100%; that point is a sanity check, not a finding. The informative region is `max_size` $< 103$, where capacity actually forces eviction decisions.

| Policy | max_size | | | | | |
|---|---|---|---|---|---|---|
| | **5** | **10** | **20** | **40** | **80** | **160** |
| *Dormant scenario* | | | | | | |
| FIFO | 0.0% | 0.0% | 0.0% | 0.0% | 0.0% | 100.0% |
| LRU | 0.0% | 0.0% | 0.0% | 0.0% | 0.0% | 100.0% |
| Importance-weighted | **100.0%** | **100.0%** | **100.0%** | **100.0%** | **100.0%** | 100.0% |
| *Revisited scenario* | | | | | | |
| FIFO | 0.0% | 0.0% | 0.0% | 0.0% | 0.0% | 100.0% |
| LRU | 0.0% | 33.3% | 33.3% | 33.3% | 33.3% | 100.0% |
| Importance-weighted | **100.0%** | **100.0%** | **100.0%** | **100.0%** | **100.0%** | 100.0% |

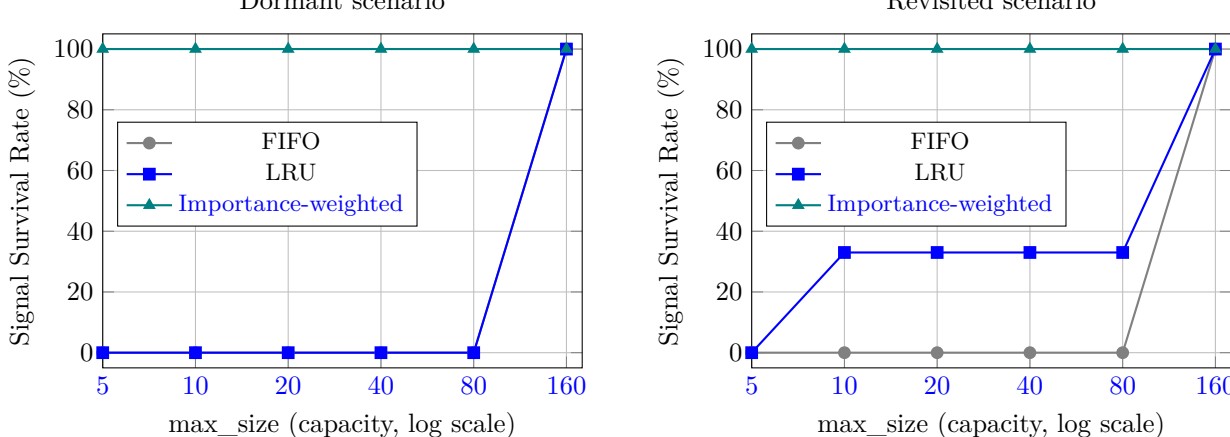

Figure 9: Signal survival rate versus memory capacity for three retention policies under dormant (left) and revisited (right) scenarios. Importance-weighted retention preserves the target signal across all tested capacities, whereas FIFO and dormant LRU retain it only once memory capacity is sufficiently large to prevent eviction. In the revisited scenario, LRU partially recovers to 33.3% survival after target revisitation, but remains below importance-weighted retention.

Two patterns are worth noting. First, *recency-based eviction is capacity-fragile regardless of variant*: FIFO loses the target signal at every tested capacity in both scenarios, and dormant LRU behaves identically to FIFO, both reach 100% survival only once `max_size` $\geq 103$, i.e., once eviction stops happening altogether, which is a convergence artifact rather than a genuine capacity effect (see the note in Table 11's caption). Second, *a single, cheap revisit changes LRU's behavior but importance-weighting still dominates*: refreshing the target's recency once partway through the distractor stream raises LRU's survival to 33.3% for `max_size` $\geq 10$ (1 of the 3 target signals survives: the single mid-stream query refreshes only the recency of the revisited signal under LRU, leaving the other two signals unprotected and evicted), versus 0% for FIFO under the same conditions, because LRU (unlike FIFO) can be "rescued" by a single query touch. Importance-weighted retention needs no such rescue: it retains the target at 100% survival across every tested capacity in both scenarios, because retention is keyed to similarity-weighted relevance rather than recency alone.

Together, these two ablations sharpen the component-attribution picture from Table 7: the multi-agent gains come from role-*separated reasoning* (prompt-level specialization plus the Critic's write-restriction)

rather than from role-conditioned memory-*retrieval* bias, and if VECTOREPISODICMEMORY is ever extended from cap-not-evict (Sec. 3.4) to an active eviction policy at scale, importance-weighted retention would be markedly more robust than FIFO or LRU under capacity pressure. The one remaining unmeasured cell from the original factorial analysis, temporal memory combined with the multi-agent architecture, is discussed as a limitation below.

## 6.8 Broader Impact

This work studies how language agents reuse past experience through persistent episodic memory. Persistent agent memory introduces potential privacy and retention risks: if an agent stores interaction content across sessions, sensitive information (user preferences, error-revealing inputs, or proprietary code snippets) may persist longer than intended. Practical mitigations include per-task or per-session memory resets (as implemented in this work), bounded retention windows, content-level sanitization before storage, and user-controllable deletion policies. Our architecture stores only natural-language failure reflections derived from test outcomes, not raw inputs, task content, or user data, so the direct risk is low in our evaluated setting. However, deployments that extend episodic memory to conversational or personal-data contexts should apply appropriate retention controls. Because MULTIAGENTREFLEXION increases LLM call volume, it also has energy and carbon costs; practitioners should weigh accuracy gains against latency, monetary costs, and energy overhead when choosing between single-agent and multi-agent deployment. An additional risk specific to reflection-based memory is stale or incorrect reasoning: if an agent stored a faulty reasoning pattern early in deployment, that pattern may continue to be retrieved and reinforce errors in future tasks. Retrieval-based memory systems in safety-critical deployments should therefore incorporate quality scoring before storage or periodic staleness audits to prevent degraded reflections from persisting indefinitely.

## 6.9 Limitations

**Trial count:** We ran 3 independent trials on the long-horizon benchmark and observed no variation in either condition. We are confident that this result is correct rather than a sampling anomaly: temporal memory's task-level failure is deterministic because a size-10 buffer with 9 distractors leaves only one Session 1 reflection, while both Session 1 reflections are required for Session 5 success, and vector memory's success is also deterministic (the Session 1 reflections consistently outscore the distractors by cosine similarity). However, conducting only 3 trials provides limited evidence for a claim of zero variance, and additional trials would strengthen the empirical basis of this result. More broadly, the limited trial count constrains inference about variance in related settings where outcomes may not be fully determined by architecture alone. In such settings, observed zero variance over 3 runs should be treated as a preliminary finding rather than a definitive stability claim.

**Single model:** All agents use Gemini 2.5 Flash. A Critic implemented with a model specialized for code review may yield further gains; this remains an open direction.

**Single model generalizability:** The results reported in this paper are specific to Gemini 2.5 Flash and may not generalize across LLM families. The magnitude of the multi-agent gain depends on the strength of the Generator and the adversarial reasoning capability of the Critic; a weaker generator model may benefit more from external critique, while a stronger one may show smaller marginal gains. Similarly, the vector memory improvement of +3.7 pp on HumanEval may vary across models with different patterns of context utilization. We make no claim that the architectural benefits reported here, particularly the +7.9 pp Pass@3 gain for MULTIAGENTREFLEXION—generalize to other LLM families. Cross-model evaluation (e.g., with GPT-4o, Claude 3.5 Sonnet, or Llama 3 70B) is an important direction for future work and is required before broad deployment recommendations can be made.

**Benchmark scope:** HumanEval tasks are semantically independent and relatively short. Extensions to multi-file refactoring and to AlfWorld (Shinn et al., 2023) (sequential decision-making) would test the robustness of both proposed extensions in more realistic deployment settings.

**Natural task-distribution evidence:** The strongest evidence for VECTOREPISODICMEMORY comes from a synthetic benchmark purpose-built to stress-test FIFO eviction, where the 100 pp Session-5 task-success improvement is guaranteed by construction. The +3.7 pp gain on HumanEval is real but modest ($d = 0.127$) and does not establish broad practical utility across all deployment scenarios. Further evaluation of natural temporally correlated workflows, such as multi-session code refactoring, personalized tutoring, or iterative document editing, is needed to determine whether semantic retrieval provides meaningful gains beyond adversarially structured long-horizon settings. Until such evidence exists, claims about the general practical utility of VECTOREPISODICMEMORY should be interpreted cautiously.

**Factorial ablation:** The role-conditioned-memory and retention-policy ablations in Sec. 6.7 narrow, but do not close, the remaining component-attribution gap identified by reviewers. A complete 2×2 ablation crossing memory type (TEMPORALMEMORY vs. VECTOREPISODICMEMORY) with agent structure (single-agent vs. MULTIAGENTREFLEXION) would cleanly separate the two contributions. The current paper evaluates MODULARBASELINE (temporal, single), VECTORREFLEXION (vector, single), and MULTIAGENTRE-FLEXION (vector, multi), leaving the temporal-memory multi-agent cell unmeasured. This missing cell would establish whether role separation provides gains independently of memory type, or whether the two interact. We leave this as an explicit direction for future work.

**Memory cap behavior:** Once the pool reaches `max_size`, it stops accepting new entries rather than evicting old ones. For deployments that run for thousands of trials, this will eventually matter. Compression strategies, clustering similar reflections, and summarising them into a single representative entry are an interesting direction we have not yet explored. Sec. 6.7 evaluates alternative retention policies, FIFO, LRU, and importance-weighted eviction, in a controlled capacity-sweep setting once eviction is actually forced; importance-weighted retention is the most robust of the three under capacity pressure and is the natural candidate should VECTOREPISODICMEMORY be extended with active eviction at scale.

## 7 Artifact Availability and Reproducibility

**Code:** The full implementation, including `VectorEpisodicMemory`, `MultiAgentReflexionAgent`, the long-horizon benchmark generator, and all evaluation scripts, is available in the attached artifacts. The artifact code package includes a `config.json.template` for API key configuration and a `README.md` with step-by-step reproduction instructions.

**Hyperparameters:** Table 12 provides a complete listing of all hyperparameters used across all agents and experiments. No hyperparameter search was performed; all values were set a priori based on REFLEXION paper defaults where applicable.

**Statistical testing:** Each agent comparison uses 164 paired binary indicators (1 if solved within $T$ trials, 0 otherwise). We use paired $t$-tests on these indicators and report 95% Wilson score confidence intervals (`statsmodels.stats.proportion.proportion_confint,method='wilson'`). Cohen's $d$ is computed on the same paired differences using the pooled standard deviation.

**Long-horizon benchmark:** The benchmark is fully procedural and deterministic given the fixed task sequence. It is structured into 5 sessions and 13 tasks total, designed so that the memory architecture, not task difficulty, determines whether an agent succeeds.

The full task prompts and test suites for all 13 tasks are included in the released code repository under `experiments/extension1_vector_memory/long_horizon_benchmark.py`.

Table 12: Complete hyperparameter listing for all agents and experiments. Values marked [†] are inherited from the original REFLEXION implementation. [‡]Memory is reset per task; the maximum intra-task pool depth is 3 trials × 3 agents = 9 entries. The default `max_size`=1,000 is never approached in practice and was used for all experiments.

| Component | Parameter | Value |
|---|---|---|
| LLM (all agents) | Model | gemini-2.5-flash |
| | Temperature | 0.7 |
| | Max tokens | 2048 |
| | Inter-req. delay | 0.5 s |
| Exponential backoff | Max retries | 5 |
| | Initial delay | 5 s |
| | Backoff factor | 2.5× |
| | Max delay | 120 s |
| TEMPORALMEMORY | Buffer size[†] | 10 |
| | Top-$k$ retrieved[†] | 3 |
| | Eviction policy | FIFO |
| VECTOREPISODICMEMORY | Encoder | all-MiniLM-L6-v2 |
| | Embedding dim | 384 |
| | Max pool size[‡] | 1,000 |
| | Top-$k$ retrieved | 5 |
| MULTIAGENTREFLEXION | Agents | Generator, Critic, Verifier |
| | Top-$k$ per agent | 3 |
| HumanEval eval. | Max trials[†] | 3 |
| | Subprocess timeout | 10 s |
| Long-horizon bench. | Sessions | 5 |
| | Total tasks | 13 |
| | Distractor tasks | 9 |
| | Independent trials | 3 |

## 8 Conclusion

Two targeted changes to REFLEXION yield complementary improvements whose strength depends on the setting. Replacing the FIFO memory with VECTOREPISODICMEMORY costs only ∼14 ms per retrieval. It stays flat up to 50,000 entries, while turning deterministic 0% Session-5 task success into 100% on our controlled long-horizon diagnostic benchmark. This task-success result is scoped to our controlled long-horizon diagnostic benchmark; on naturally distributed coding tasks (HumanEval), the gain is modest (+3.7 pp, $d = 0.127$), confirming that semantic retrieval is most valuable when task sessions are temporally correlated. On HumanEval, it adds 3.7 pp ($d = 0.127$). The Generator–Critic–Verifier architecture adds 7.9 pp Pass@3 ($d = 0.301$) and 12.2 pp Pass@1 over the modular baseline, with average trials dropping to 1.03 at a cost overhead of roughly $0.0005 per task. Two additional component-attribution ablations (Sec. 6.7) refine this picture: role-conditioning the shared memory pool's retrieval does not add a statistically detectable lift beyond plain semantic retrieval ($p = 1.0000$), so the multi-agent gains are attributable to role-*separated reasoning* rather than role-conditioned *memory retrieval*; and among retention policies evaluated under forced eviction, importance-weighted retention is markedly more robust to capacity pressure than FIFO or LRU. The lesson we take from this is practical: match the fix to the failure mode. When tasks are correlated across sessions, semantic memory is highly beneficial. When tasks are independent but reasoning is shallow, role separation helps more than memory. The obvious next experiment would be to combine both extensions and give each Agent its own VECTOREPISODICMEMORY pool, which we intend to pursue, especially for AlfWorld and multi-file refactoring, where both problems are likely to matter simultaneously.

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
