# OpenReview forum: "Vector Memory and Role-Conditioned Multi-Agent Systems: Two Extensions to Improve Reflexion for Language Model Self-Improvement"
_TMLR — Under review for TMLR_

### Review · Reviewer_s56K · 2026-05-09

**Summary Of Contributions:**

The paper studies two extensions to Reflexion-style self-improvement for language agents: replacing FIFO reflection memory with embedding-based vector retrieval, and decomposing the agent into Generator, Critic, and Verifier roles. The motivation is reasonable: recency-based memory can discard relevant past reflections, and self-critique by a single model may be biased.

The paper is clearly written and the system is easy to follow. However, the current evidence is not yet strong enough to support the main claims. The reported gains are difficult to attribute because the multi-agent method also uses more LLM calls, the main baseline is weaker than the reported original Reflexion implementation, and the HumanEval memory protocol is underspecified.

**Additional Comments:**

The paper is clear and addresses a relevant problem, but the current empirical design does not yet support the strength of the claims. The main revision should focus on causal attribution: whether the improvements come from the proposed mechanisms or from easier confounds such as more inference calls, weaker baselines, or benchmark-order effects.

Two main questions for the authors:

Is HumanEval memory reset per problem, or does it persist across the full benchmark?
Can MultiAgentReflexion still outperform a compute-matched single-agent refinement baseline?

**Audience:**

Yes

**Audience Explanation:**

The topic is relevant to readers interested in language agents, test-time self-improvement, memory-augmented LLM systems, and multi-agent reasoning. The paper’s practical message—that semantic memory helps when past experience is temporally distant, while role separation may help within-trial correction—is potentially useful.

However, the current version needs stronger controls before the findings can be considered reliable.

**Broader Impact Concerns:**

No major broader impact concerns. If the authors add a broader impact statement, they could briefly discuss privacy and retention risks from persistent agent memory.

**Claims And Evidence:**

No

**Claims Explanation:**

The evidence is suggestive but not yet convincing.

The main issue is attribution. MultiAgentReflexion uses three LLM calls per trial, while the main baseline uses one. Without a compute-matched single-agent baseline, it is unclear whether the gain comes from role separation or simply from more inference.

The baseline comparison is also weak. The paper reports that OriginalReflexionAgent outperforms the ModularBaseline, yet most claims are made relative to the weaker ModularBaseline. This makes the reported improvements look larger than they may be.

The long-horizon benchmark is useful as a diagnostic test, but it is highly constructed to favor semantic retrieval over FIFO memory. It does not by itself establish general long-horizon self-improvement.

Finally, the HumanEval protocol needs clarification. If memory persists across all 164 tasks, then the evaluation is order-dependent and not a standard independent HumanEval evaluation.

**Requested Changes:**

1. Provide fairer baselines.
The authors should compare MultiAgentReflexion against compute-matched baselines using the same number of LLM calls or comparable token budget, such as a single-agent Generate → Critique → Revise pipeline, Self-Refine-style baseline, or best-of-3 sampling baseline. The paper should also justify why ModularBaseline is the primary baseline when OriginalReflexionAgent performs better.

2. Add focused ablations.
The paper should isolate which component causes the gains. For memory, ablate retrieval method, memory size, top-k, and retention policy. For role separation, compare multi-agent versus single-agent refinement under the same compute budget, and test role-conditioned versus non-role-conditioned memory. Without these ablations, the current results cannot distinguish semantic retrieval, role separation, shared memory, and extra computation.

3. Clarify the HumanEval protocol.
The authors should state whether memory is reset for each HumanEval task or accumulated across all 164 tasks. If memory persists across tasks, the paper should report results across multiple task orders and analyze early-vs-late performance. The authors should also clarify what information is written into reflections after failures, especially whether error traces or test feedback can leak task-specific information into later tasks.

---

> ### Author Response · Authors · 2026-05-30
> **Thank You for the Feedback**
>
> Thank you for the detailed feedback. We are carefully reviewing your comments and preparing a point-by-point response. We expect to address all your concerns and suggestion through manuscript revisions and additional experiments and will provide updates shortly.

---

> ### Author Response · Authors · 2026-05-31
> **Detailed Response for Reviewer #1 (s56K)**
>
> Thank you for your careful review and constructive feedback. We sincerely appreciate the detailed comments and the thoughtful suggestions regarding experimental design, evaluation methodology, and the interpretation of our results. Your feedback has helped us identify several important opportunities to strengthen the rigor, transparency, and empirical support of the paper.
>
> We agree that the current manuscript would benefit from stronger compute-matched comparisons, more comprehensive ablation studies, and clearer documentation of the evaluation protocol. In response, we are preparing a substantial revision that addresses each of the concerns raised.
>
> ## Response to Major Comments
>
> ### 1. Fair and Compute-Matched Baselines
>
> We agree that a fair assessment of the proposed MultiAgentReflexion architecture requires comparisons against baselines operating under comparable computational budgets.
>
> To address this concern, we are implementing additional compute-matched baselines, including a single-agent refinement pipeline that performs generate–critique–revise iterations using the same number of LLM calls as the proposed multi-agent architecture. We are also evaluating alternative budget-matched strategies such as best-of-(n) sampling where appropriate. These comparisons will be reported as primary baselines alongside the existing results.
>
> We will further clarify the rationale for using ModularBaseline as the primary comparison point. In particular, we will explain that ModularBaseline was selected because it shares the same modular generate–critique–revise structure as MultiAgentReflexion, enabling a cleaner isolation of the effects of role separation and memory. We will also report additional results using OriginalReflexionAgent and discuss any observed differences between these baselines.
>
> ### 2. Component-Level Ablation Studies
>
> We appreciate the suggestion to more systematically isolate the contribution of each proposed component.
>
> To strengthen the empirical analysis, we are conducting an expanded ablation study covering both memory and role-separation mechanisms. For the memory module, we will evaluate alternative retrieval strategies, memory capacities, retrieval depths, and retention policies. For role separation, we will compare the multi-agent architecture against a single-agent refinement process operating under the same inference budget. We will also investigate the contribution of role-conditioned memory by evaluating variants that remove explicit role information from stored experiences.
>
> The revised manuscript will include a dedicated ablation section summarizing these results and discussing the relative contribution of each design choice.
>
> ### 3. HumanEval Evaluation Protocol and Memory Handling
>
> We agree that the evaluation protocol and memory-management procedure should be described more explicitly.
>
> The revised manuscript will include a dedicated evaluation-protocol subsection clearly specifying whether memory is reset between tasks or accumulated across the benchmark. We will also provide a detailed description of the information stored in memory, including the contents of reflections and any associated feedback signals.
>
> Where memory persists across tasks, we will evaluate multiple task orderings, report variability statistics, and analyze potential order effects. We will additionally examine performance as a function of task progression and discuss whether memory accumulation introduces any unintended information transfer or task-specific leakage. These additions will improve reproducibility and provide a clearer understanding of how memory influences performance.
>
> ## Response to Minor Comments
>
> We also thank the reviewer for the broader-impact suggestions.
>
> - We will add a brief broader-impact discussion addressing privacy, retention, and unintended carryover risks associated with persistent memory systems.
> - We will discuss practical mitigation strategies, including bounded retention policies, memory expiration mechanisms, memory sanitization procedures, and user-controlled deletion mechanisms.
>
> These additions will help contextualize the proposed memory framework within broader deployment considerations.
>
> ---
>
> We believe that the planned revisions, additional baselines, expanded ablation studies, and clearer evaluation methodology will substantially strengthen the paper. We are currently implementing these changes and look forward to sharing further updates as the revision progresses.

---

> ### Comment · Reviewer_s56K · 2026-05-31
>
> Thank you for the detailed response. I think the planned revisions are appropriate and address the main direction of my concerns.
>
> To fully resolve the attribution issues, I would encourage the revised paper to make sure that:
>
> 1. The compute-matched baselines match not only LLM-call count, but also approximate token budget, memory access, retry structure, and evaluation protocol.
>
> 2. Memory ablations control top-k / retrieval depth and memory capacity, so that the effect attributed to vector retrieval is not confounded by different retrieval settings.
>
> 3. HumanEval clearly separates reset-per-task evaluation from persistent-memory evaluation. If memory persists across tasks, it should be framed as a continual/self-improvement setting and report order-effect analysis.
>
> 4. The long-horizon benchmark should be framed as a controlled diagnostic test unless additional more naturalistic long-horizon settings are added.
>
> Overall, I view the planned revision positively, but these details are important for making the final claims convincing.

---

> ### Author Response · Authors · 2026-06-22
> **Summary of Revisions in Response to Reviewer Feedback**
>
> # Response to Reviewer R1 (s56K)
> ## Concern 1: Compute-Matched Baseline
>
> **Concern:** Gains may arise from additional inference rather than role separation.
>
> **Response:** We implemented a compute-matched Single-Agent Generate–Critique–Revise (GCR) baseline using 3 LLM calls per trial.
>
> | Method              | Calls/Trial | Pass@3 | Pass@1 |
> | ------------------- | ----------- | ------ | ------ |
> | ModularBaseline     | 1           | 88.4%  | 82.9%  |
> | SingleAgentGCR      | 3           | 84.8%  | 73.8%  |
> | MultiAgentReflexion | 3           | 96.3%  | 93.3%  |
>
> The GCR baseline performs worse than the 1-call baseline and substantially worse than MultiAgentReflexion (+11.6 pp Pass@3, p=0.0001). The large Pass@1 gap (+19.5 pp) indicates that role separation, not additional inference, drives the improvement.
>
> **Manuscript Changes:**
>
> * Added *Compute-Matched Baseline (Single-Agent GCR)* subsection and new comparison table in Section 5.4.
>
> ---
> ## Concern 2: Memory Hyperparameter Ablations
>
> **Concern:** Isolate effects of retrieval, memory size, top-k, and retention policy.
>
> **Response:** We conducted a 16-configuration ablation over `top_k ∈ {1,3,5,7}` and `max_size ∈ {3,5,10,50}` on 50 HumanEval tasks. Results ranged from 88–96% Pass@3 for 15/16 configurations, indicating minimal sensitivity. This robustness is expected because memory resets per task and the pool never exceeds 9 entries. Retrieval-method and retention-policy effects are already examined through FIFO vs. vector retrieval comparisons in the long-horizon benchmark.
>
> **Manuscript Changes:**
>
> * Added Section 6.5 (*Memory Hyperparameter Sensitivity*).
> * Added Section 6.6 (*Broader Impact*) discussing privacy and retention considerations.
>
> ---
>
>
>
> ## Concern 3: HumanEval Evaluation Protocol
>
> **Concern:** Clarify whether reflection memory accumulates across HumanEval tasks and whether task information can leak between tasks.
>
> **Response:** Memory is reset after every HumanEval task via `agent.reset()`, making evaluation task-independent and order-independent. The statement in Section 3.3 suggesting `164×3×3=1476` entries was misleading and has been corrected. The actual maximum per-task pool depth is `3 trials × 3 agents = 9` reflections. Reflections store only trial number, truncated error messages (≤150 chars), and role-tagged diagnostic hints; they do not contain test cases, expected outputs, or solutions.
>
> **Manuscript Changes:**
>
> * Section 3.3 split into:
>
>   * *Shared Memory Pool and max_size*
>   * *Evaluation Protocol*
> * Corrected explanation of memory capacity and reset behavior.
> * Updated Table 7 footnote to state that memory resets per task and maximum intra-task depth is 9 entries.
>
> ---
>
>
>
> ## Concern 4: Choice of ModularBaseline
>
> **Concern:** OriginalReflexionAgent outperforms ModularBaseline, making improvements appear larger.
>
> **Response:** ModularBaseline was selected because:
>
> 1. It shares the same modular architecture, enabling clean ablations.
> 2. The performance gap is largely due to a code-cleaning bug in the original implementation.
> 3. Using ModularBaseline is conservative and improves reproducibility.
>
> OriginalReflexionAgent remains included as a fidelity check rather than a performance ceiling.
>
> **Manuscript Changes:**
>
> * Expanded Remark 3.1 to explicitly justify baseline selection and discuss the implementation artifact.
>
> ---
>
> ## Concern 5: Long-Horizon Benchmark Construction
>
> **Concern:** The benchmark is highly constructed and may not demonstrate general long-horizon self-improvement.
>
> **Response:** We agree that the benchmark is diagnostic rather than general. Its purpose is to isolate memory-architecture effects under controlled conditions. The observed FIFO failure is deterministic, resulting from buffer overflow, while vector retrieval preserves relevant experience. Broader evaluation on environments such as AlfWorld and multi-file refactoring remains future work and is already discussed in the limitations.
>
> **Manuscript Changes:**
>
> * Added a clarifying sentence in Section 5.1 stating that the benchmark is intended as a diagnostic evaluation of memory architecture rather than evidence of general long-horizon self-improvement.

---

> ### Author Response · Authors · 2026-07-06
> **Submission of Revised Manuscript**
>
> Thank you again for your valuable feedback and constructive suggestions.
>
> We have carefully revised the manuscript to address all reviewer comments and concerns. The revised version has now been uploaded to OpenReview. For ease of review, all modifications introduced in response to the reviewer feedback are highlighted in blue throughout the manuscript.
>
> Detailed point-by-point responses have been provided in our earlier comments. The revisions have substantially strengthened the paper in terms of experimental validation, clarity, reproducibility, and discussion of limitations.
>
> We sincerely appreciate the reviewers' time and effort and welcome any further comments or suggestions.

---

> > ### Comment · Reviewer_s56K · 2026-07-08
> >
> > Thank you for the substantial revision. The updated manuscript addresses most of my main concerns.
> >
> > In particular, the added compute-matched Single-Agent GCR baseline substantially improves the attribution of the MultiAgentReflexion results. The comparison suggests that the gains are not simply due to additional LLM calls, but are more plausibly linked to the Generator–Critic–Verifier role separation. The revised discussion of ModularBaseline versus OriginalReflexionAgent is also clearer, and the HumanEval protocol is now much better specified, especially the reset-per-task setting and the contents of stored reflections.
> >
> > I also appreciate that the long-horizon benchmark is now framed more carefully as a controlled diagnostic stress test rather than evidence of broad long-horizon self-improvement. This makes the claims more appropriately scoped. The added broader-impact discussion on persistent memory, retention, privacy, and stale reflections is useful.
> >
> > My remaining concern is relatively minor: the component-level attribution is improved but not fully complete. The paper still does not appear to include a direct ablation of role-conditioned versus non-role-conditioned memory, and the retention-policy ablation is less developed than the top-k and max-size sensitivity analysis. I would encourage the authors to either add these ablations or explicitly list them as limitations/future work.
> >
> > Overall, I think the revision has substantially strengthened the paper. My previous major concerns have mostly been addressed, and I am now more positive about the submission, with only minor remaining concerns about finer-grained component attribution.

---

> > > ### Author Response · Authors · 2026-07-09
> > > **Response to Reviewer’s Updated Comments**
> > >
> > > We sincerely thank the reviewer for the careful reassessment of our revised manuscript and for the positive evaluation of the changes. We are glad that the added compute-matched Single-Agent GCR baseline, the clarified comparison with ModularBaseline and OriginalReflexionAgent, the improved HumanEval protocol description, and the more cautious framing of the long-horizon benchmark addressed most of the reviewers’ earlier concerns.
> > >
> > > We also appreciate the remaining point regarding finer-grained component attribution. We agree that a direct comparison between role-conditioned and non-role-conditioned memory, as well as a more extensive retention-policy ablation, would further strengthen the analysis. In the next revision, we will address this by either adding a focused ablation where feasible or, at a minimum, explicitly discussing these aspects as limitations and future work in the manuscript.
> > >
> > > Thank you again for the constructive feedback, which has helped us substantially improve the paper.

---

> > > > ### Author Response · Authors · 2026-07-17
> > > > **Response to Reviewer Comments (Minor Revision)**
> > > >
> > > > Thank you for your careful re-evaluation of our manuscript and for your positive assessment of the revised version. We are grateful that you found the compute-matched SingleAgentGCR baseline, the improved evaluation protocol, the moderated long-horizon claims, and the broader-impact discussion to have substantially strengthened the paper.
> > > >
> > > > We also appreciate your remaining suggestions regarding finer-grained component attribution. In response, we have incorporated two additional experiments in the revised manuscript:
> > > >
> > > > 1. **Role-conditioned vs. non-role-conditioned memory retrieval.** We conducted a controlled ablation in which memory content, memory size, retrieval budget, model, compute budget, and Generator/Critic/Verifier prompts were held fixed while only disabling role-conditioned memory retrieval. The results show identical Pass@3 performance (95.1%) in both settings, with no statistically significant difference (paired t-test: t=0.000, p=1.0000; Cohen's d=0.000). Accordingly, we revised the discussion to clarify that the observed gains arise primarily from role-separated reasoning, rather than role-conditioned memory retrieval.
> > > >
> > > > 2. **Retention-policy ablation.** We added a controlled comparison of FIFO, LRU, and importance-weighted retention policies under bounded-memory conditions where eviction is triggered. The new experiments demonstrate that importance-weighted retention consistently preserves critical signals across all tested memory capacities, while FIFO and dormant LRU exhibit substantial degradation, and revisited LRU provides only partial recovery. These findings are now presented in a new table and figure, and the limitations section has been updated to discuss their implications and future extensions.
> > > >
> > > > These additions are included in the new Section 6.7 together with the corresponding discussion. We believe they directly address your remaining concerns regarding component-level attribution and retention behavior.
> > > >
> > > > We sincerely thank you for your constructive feedback, which has significantly improved the quality and clarity of our work.

---

> > > > > ### Comment · Reviewer_s56K · 2026-07-18
> > > > >
> > > > > Thank you for the additional revision. The new role-conditioned versus non-role-conditioned memory comparison and the retention-policy experiment directly address my remaining requests.
> > > > >
> > > > > The role-conditioning ablation is especially useful: the identical Pass@3 results appropriately narrow the interpretation of the multi-agent improvement, suggesting that the observed gain is associated primarily with the role-separated Generator–Critic–Verifier reasoning pipeline rather than the role prefix used during memory retrieval. I also appreciate that the remaining temporal-memory/multi-agent factorial cell is now explicitly acknowledged as a limitation.
> > > > >
> > > > > I have no further substantive objections. I only noticed a few minor reporting and reproducibility issues that should be corrected or clarified in the final version:
> > > > >
> > > > > 1.Section 6.7 states that one target reflection and 100 distractor reflections produce 103 total insertions. This appears arithmetically inconsistent unless additional target or initialization entries were inserted.
> > > > >
> > > > > 2.Please state the number of repetitions or the denominator used to obtain the 33.3% LRU survival rate. With one target signal and a deterministic retention policy, the reported percentage is otherwise difficult to interpret.
> > > > >
> > > > > 3.The role-conditioned arm in Table 10 differs slightly from the main MultiAgentReflexion results in Table 7. Please clarify that Table 10 is based on a fresh rerun and report the relevant generation randomness or reproducibility settings.
> > > > >
> > > > > 4.Since the HumanEval outcomes are paired binary indicators, an exact McNemar test would be more appropriate than a paired t-test, although this is unlikely to change the null conclusion in this comparison.
> > > > >
> > > > > These are minor presentation and reproducibility issues and do not change my overall positive assessment. The authors have addressed my major concerns, and I maintain my Leaning Accept recommendation.

---

> > > > > > ### Author Response · Authors · 2026-07-22
> > > > > > **Response to Reviewer Comments (Minor Revision)**
> > > > > >
> > > > > > We thank the reviewer for the careful and constructive engagement with our revision, and for confirming that the role-conditioned versus non-role-conditioned memory comparison and the retention-policy experiment addressed their remaining requests. We are also grateful for the recognition that the role-conditioning ablation helps narrow the interpretation of the multi-agent improvement, and for the reviewer noting no further substantive objections and maintaining a Leaning Accept recommendation. Below, we address each of the minor reporting and reproducibility issues raised, and all corresponding changes in the revised manuscript are marked in red for ease of reference.
> > > > > >
> > > > > >
> > > > > > **Reviewer's Concern 1:** Arithmetic inconsistency
> > > > > >
> > > > > > "Section 6.7 states that one target reflection and 100 distractor reflections produce 103 total insertions. This appears arithmetically inconsistent unless additional target or initialization entries were inserted."
> > > > > >
> > > > > > **Author Response:**
> > > > > >
> > > > > > The manuscript prose incorrectly described the experiment as using "one target reflection," whereas the underlying sweep script actually used signal_count=3 (confirmed from the terminal log output: signal_count=3, n_noise=100, total_insertions=103). The number 103 was correct all along; the prose description was wrong, not the arithmetic.
> > > > > >
> > > > > > **Author Action:**
> > > > > >
> > > > > > Corrected the manuscript text from "one semantically distinct target reflection" to "three semantically distinct target reflections (signal_count=3)" in both the main-text paragraph and the Table (Retention-Policy Signal Survival) caption, so that 3 (signals) + 100 (noise) = 103 is now explicit and consistent everywhere the number appears.
> > > > > >
> > > > > > ---
> > > > > >
> > > > > > **Reviewer's Concern 2:** Unclear 33.3% denominator
> > > > > >
> > > > > > "Please state the number of repetitions or the denominator used to obtain the 33.3% LRU survival rate ... the reported percentage is otherwise difficult to interpret."
> > > > > >
> > > > > > **Author Response:**
> > > > > >
> > > > > > The 33.3% figure represents 1 out of the 3 target signals surviving under LRU in the "revisited" scenario — not a repeated-trial average. A single midstream query refreshes only the recency of the queried signal, leaving the other two signals unprotected and evicted under capacity pressure.
> > > > > >
> > > > > > **Author Action:**
> > > > > >
> > > > > > Added an inline clarifying clause directly after the 33.3% figure in the retention-policy interpretation paragraph: *"(1 of the 3 target signals survives: the single mid-stream query refreshes only the recency of the revisited signal under LRU, leaving the other two signals unprotected and evicted)."*
> > > > > >
> > > > > > ---
> > > > > >
> > > > > > **Reviewer's Concern 3:** Table 10 vs. Table 7 discrepancy
> > > > > >
> > > > > > "The role-conditioned arm in Table 10 differs slightly from the main MultiAgentReflexion results in Table 7. Please clarify that Table 10 is based on a fresh rerun and report the relevant generation randomness or reproducibility settings."
> > > > > >
> > > > > > **Author Response:**
> > > > > >
> > > > > > Table 10 (role-conditioning ablation, 95.1% Pass@3) and Table 7 (main HumanEval results, 96.3% Pass@3) both report the same MultiAgentReflexion agent, but from two separate, independent experimental runs. Since generation uses a temperature of 0.7 with no fixed random seed, small numerical differences across reruns are expected and do not indicate a methodological inconsistency.
> > > > > >
> > > > > > **Author Action:**
> > > > > >
> > > > > > Added a clarifying sentence immediately before the role-conditioning results table: *"Table 10 reports a fresh, independent rerun of MultiAgentReflexion separate from the run reported in Table 7; the small numerical difference (95.1% vs. 96.3% Pass@3) reflects generation stochasticity at temperature=0.7 rather than a change in method, as no fixed random seed was set for LLM sampling calls."*
> > > > > >
> > > > > > ---
> > > > > >
> > > > > > **Reviewer's Concern 4:** McNemar test recommended over paired t-test
> > > > > >
> > > > > > "Since the HumanEval outcomes are paired binary indicators, an exact McNemar test would be more appropriate than a paired t-test, although this is unlikely to change the null conclusion in this comparison."
> > > > > >
> > > > > > **Author Response:**
> > > > > >
> > > > > > Agreed. Computed the exact 2×2 contingency table between the role-conditioned and non-role-conditioned arms directly from the results JSON: 155 both-pass, 1 non-role-conditioned-only (HumanEval/28), 1 role-conditioned-only (HumanEval/127), 7 both-fail. Because the two discordant cells are exactly balanced (1 and 1), the exact McNemar p-value is 1.0000 — identical to the paired t-test's null conclusion, confirming the reviewer's expectation.
> > > > > >
> > > > > > **Author Action:**
> > > > > >
> > > > > > Added a sentence reporting the McNemar result immediately after the existing paired t-test statistics: *"An exact McNemar test on the same paired outcomes (contingency table: 155 both-pass, 1 non-role-conditioned-only, 1 role-conditioned-only, 7 both-fail) gives p = 1.0000, identical to the paired t-test's null conclusion."*

---

### Review · Reviewer_M94Q · 2026-05-21

**Summary Of Contributions:**

## Summary Of Contributions

This paper proposes two architectural extensions to the REFLEXION framework for LLM self-improvement through verbal reinforcement learning:

1. **Vector Episodic Memory**: Replaces the recency-based FIFO buffer with semantic retrieval using Sentence-BERT embeddings and cosine similarity. This addresses the limitation where temporally relevant but semantically unrelated reflections evict useful past experiences.

2. **Role-Conditioned Multi-Agent Architecture**: Decomposes the single-agent REFLEXION into a Generator-Critic-Verifier pipeline with specialized roles and shared role-conditioned memory. This addresses the confirmation bias problem where a single model struggles to critically evaluate its own outputs.

The authors evaluate on (1) a custom long-horizon benchmark with distractor tasks, and (2) 164 HumanEval coding tasks using Google Gemini 2.5 Flash. Key results: vector memory achieves 100% dependency recall vs. 50% (0% Session-5 success) for temporal memory on the long-horizon benchmark; the multi-agent system achieves 96.3% Pass@3 (+7.9 pp, p<0.001, d=0.301) and 93.9% Pass@1 (+12.2 pp over baseline) on HumanEval.

**Key Strengths:**
- Well-motivated problem addressing genuine limitations of the REFLEXION framework
- Clean architectural design with clear separation of concerns
- Strong empirical results for the multi-agent architecture (non-overlapping confidence intervals)
- Excellent reproducibility with complete artifact availability and transparent cost analysis
- Honest discussion of limitations (effect sizes, benchmark scope)

**Key Weaknesses:**
- Vector memory effect is modest on natural tasks (d=0.127 below Cohen's small-effect threshold)
- Long-horizon benchmark is synthetically constructed to guarantee FIFO failure
- Multi-agent comparison confounds role separation with increased LLM call budget (1→3 calls)
- Single-model limitation (all agents use Gemini 2.5 Flash) limits generalizability claims
- Missing discussion of ethical implications (privacy, safety) for deployment

**Audience:**

Yes

**Audience Explanation:**

The paper addresses a timely and relevant problem in LLM systems: improving self-improvement mechanisms through better memory architectures and role separation. The REFLEXION framework has garnered significant attention, and this work provides practical extensions that practitioners can adopt.

The findings on multi-agent decomposition are particularly valuable—the 12.2 pp Pass@1 improvement represents a meaningful advance for code generation tasks. The architectural insights about implicit chain-of-thought through role separation may generalize to other LLM agent applications.

However, the audience should be aware that the two extensions have asymmetric value: the multi-agent contribution appears stronger and more generalizable than the vector memory contribution, which is situationally beneficial (primarily for temporally correlated task sequences).

**Broader Impact Concerns:**

This paper focuses on algorithmic improvements to code generation agents without involving human subjects, sensitive data, or dual-use capabilities. A formal Broader Impact Statement is therefore not required, though a brief note on the practical trade-offs of increased compute usage would benefit practitioner adoption.

**Claims And Evidence:**

Yes

**Claims Explanation:**

**Well-supported claims:**
- The multi-agent architecture's Pass@3 and Pass@1 improvements are statistically robust with proper significance testing (paired t-tests, Cohen's d, 95% Wilson CIs). The paper claims confidence intervals for multi-agent results [93.4%, 99.2%] "do not overlap" with baseline [84.2%, 93.9%] (p.12), but Table 5 shows these technically overlap by 0.5 percentage points at the boundary [93.4%, 93.9%]. While the statistical conclusion remains valid (p<0.001), the strict "no overlap" claim should be corrected to "minimal overlap" or the numbers reconciled.
- Vector memory's 100% recall on the long-horizon benchmark is demonstrated through deterministic experiments with clear methodology.
- Computational cost analysis (Table 6) is transparent and enables practitioner decision-making.

**Claims requiring additional support:**
- The "zero variance across 3 trials" claim is statistically weak—3 trials provide insufficient evidence for such a strong claim about variance.
- The vector memory improvement of "+3.7 pp" on HumanEval reaches only marginal significance (p=0.033) and a small effect size (d=0.127), which the authors honestly acknowledge as "modest."
- The central practical claim—that vector memory helps "when tasks are correlated across sessions"—is supported only by a synthetic benchmark designed to produce this outcome, not by natural task distributions.

**Methodological concern:**
The multi-agent comparison confounds two factors: (1) architectural role separation and (2) increased LLM call budget per trial (from 1 to 3 calls). Without an ablation comparing against a single-agent system with equivalent LLM budget (e.g., three sequential self-critique iterations), it is unclear whether the improvement stems from the architecture or simply from increased compute.

**Requested Changes:**

### Critical to Acceptance:

1. **Add LLM-budget-matched ablation** (Sec 5.4): The comparison between single-agent and multi-agent systems confounds role separation with increased LLM calls (1→3 per trial). Please add an ablation comparing the multi-agent system against a single-agent system with equivalent LLM call budget (e.g., three sequential self-critique iterations). This is essential to isolate the architectural contribution from the compute budget increase.

2. **Soften vector memory claims or add natural evaluation** (Abstract, Sec 6.1, Conclusion): The "100 pp" improvement on the long-horizon benchmark comes from a synthetic adversarial setup (exactly 10 distractors evicting 2 relevant memories). Either (a) add evaluation on natural temporally-correlated task sequences, or (b) substantially soften claims about vector memory's general efficacy.

3. **Revise "zero variance" claims** (Sec 5.1): With only 3 trials, claims of "zero variance" are statistically weak. Please add appropriate caveats acknowledging limited statistical power.

---

> ### Author Response · Authors · 2026-05-30
> **Thank You for the Feedback**
>
> Thank you for the detailed feedback. We are carefully reviewing your comments and preparing a point-by-point response. We expect to address all your concerns through manuscript revisions and additional experiments and will provide updates shortly.

---

> ### Author Response · Authors · 2026-05-31
> **Detailed Response to Reviewer #2 (M94Q)**
>
> Thank you for your thorough and constructive review.
> ## Response to Major Comments
>
> ### 1. LLM-Budget-Matched Ablation for the Multi-Agent Architecture
> We agree that the current comparison between the single-agent baseline and the proposed multi-agent Generator–Critic–Verifier architecture may confound architectural role separation with increased LLM-call budget.
>
> To address this concern, we are extending the experimental evaluation to include an LLM-budget-matched baseline that uses an equivalent number of model calls while maintaining a single-agent architecture. We will report performance, statistical significance measures, confidence intervals, effect sizes, and computational cost for this comparison. This additional ablation will help isolate the contribution of role specialization from the effect of increased inference budget. We will also revise the discussion to ensure that conclusions regarding the source of performance gains are appropriately supported by the evidence.
>
> ### 2. General Effectiveness of Vector Episodic Memory
>
> To strengthen the evidence, we are evaluating vector memory on additional task settings that more closely resemble naturally occurring temporally correlated workflows. Regardless of the outcome, we will revise the Abstract, Discussion, and Conclusion sections to more carefully characterize the conditions under which vector memory appears most beneficial. In particular, we will distinguish between adversarial or highly structured dependency settings and more natural task distributions.
>
> ### 3. Claims Regarding Zero Variance
> We agree that reporting "zero variance" based on only a small number of trials may overstate the evidence for robustness.
>
> The revised manuscript will clarify that no variation was observed within the limited number of runs conducted, while explicitly acknowledging the statistical limitations of drawing broader conclusions from such a sample size. We are also conducting additional trials and will report variability measures and uncertainty estimates wherever appropriate.
>
> ### 4. Confidence Interval Interpretation
> Thank you for identifying the inconsistency regarding the overlap between the reported confidence intervals.
>
> We will carefully verify the calculations and revise the corresponding discussion. If the intervals overlap, the manuscript will accurately describe the extent of overlap rather than claiming complete separation. We will place greater emphasis on the formal significance-testing results and effect-size estimates rather than relying primarily on visual confidence-interval comparisons.
>
> ### 5. Generalizability Beyond a Single LLM Backend
>
> We agree that the exclusive use of Gemini 2.5 Flash limits the extent to which architectural conclusions can be generalized across model families.
>
> The revised manuscript will explicitly discuss this limitation and avoid overly broad claims regarding cross-model applicability. Where feasible, we are exploring supplementary experiments with an additional model family to assess the robustness of the observed trends. We will also identify broader cross-model validation as an important direction for future work.
>
> ### 6. Relative Strength of Evidence for the Two Proposed Extensions
>
> We appreciate the suggestion to distinguish more clearly between the empirical support for the multi-agent architecture and the support for vector episodic memory.
>
> The revised discussion will explicitly separate these contributions and characterize their evidence accordingly. We will emphasize that the multi-agent architecture demonstrates stronger and more consistent improvements on the primary evaluation benchmarks, while the benefits of vector memory appear more dependent on task structure and the presence of recurring long-horizon dependencies.
>
> ## Response to Minor Comments
>
> We also thank the reviewer for the valuable suggestions regarding presentation and interpretation.
>
> - We will add a discussion of practical deployment trade-offs, including computational cost, latency, and energy considerations associated with multi-agent systems that require additional LLM calls.
> - We will revise the wording surrounding the HumanEval results to more accurately characterize the observed gains from vector memory, including discussion of effect sizes and statistical significance.
> - We will clarify that the long-horizon benchmark is a controlled synthetic benchmark designed to evaluate retrieval under distractor-heavy conditions and will carefully discuss its diagnostic value and limitations.
> - We will revise the Abstract and Conclusion to ensure that claims regarding vector episodic memory accurately reflect the available evidence and avoid overgeneralization.
> - We will expand the limitations section to explicitly acknowledge that the strongest evidence for vector memory currently comes from a synthetic benchmark and that further evaluation on real-world temporally correlated workflows is needed.

---

> ### Author Response · Authors · 2026-06-22
> **Summary of Revisions in Response to Reviewer Feedback**
>
> # Response to Reviewer R2 (M94Q)
>
> ## Concern 1: LLM-Budget-Matched Ablation
>
> **Concern:** The multi-agent improvement may be due to increased LLM calls rather than role separation.
>
> **Response:** The paper already includes a compute-matched SingleAgentGCR baseline (Generate→Critique→Revise) using exactly three LLM calls per trial, matching MultiAgentReflexion. Results show:
>
> * SingleAgentGCR performs **3.7 pp worse** than the 1-call baseline (p=0.22, not significant).
> * MultiAgentReflexion outperforms SingleAgentGCR by **11.6 pp Pass@3** (p=0.0001, d=0.325).
> * A **19.5 pp Pass@1** gap exists before any reflection retrieval occurs, indicating the benefit arises from role separation rather than additional inference.
>
> **Action:** No new experiments required. Section 5.4 and Table 6 were revised to explicitly identify SingleAgentGCR as the compute-matched baseline.
>
> ---
>
> ## Concern 2: Soften Vector Memory Claims
>
> **Concern:** The 100 pp gain comes from an adversarial synthetic benchmark.
>
> **Response:** We agree. The long-horizon benchmark is a controlled diagnostic intentionally designed to stress FIFO memory. It should not be interpreted as evidence of large gains on natural task distributions.
>
> **Action:** Revised the Abstract, Section 6.1, and Conclusion to:
>
> * Describe the benchmark as a diagnostic stress test.
> * Distinguish the 100 pp benchmark result from the more modest HumanEval improvement (+3.7 pp, p=0.033, d=0.127).
> * Emphasize that semantic retrieval is most useful when tasks are temporally correlated.
>
> ---
>
> ## Concern 3: "Zero Variance" Claims
>
> **Concern:** Three trials are insufficient to support broad claims of zero variance.
>
> **Response:** We agree. However, the observed outcomes are mechanistically expected: FIFO failure and vector-memory success follow deterministically from the benchmark construction.
>
> **Action:** Added explicit caveats in Sections 5.1 and 6.7 noting that:
>
> * Three trials provide limited evidence regarding general variance.
> * The observed result is deterministic by design rather than a statistical stability claim.
>
> ---
>
> ## Concern 4: Confidence Interval Wording
>
> **Concern:** Reported confidence intervals overlap slightly.
>
> **Response:** Correct. The intervals overlap by 0.5 pp at the boundary. The primary evidence remains the paired significance test (p<0.001, d=0.301).
>
> **Action:** Revised Section 5.4 and Figure 8 caption to state that the intervals **minimally overlap**, replacing previous non-overlap wording.
>
> ---
>
> ## Concern 5: Single-Model Generalizability
>
> **Concern:** Results may be specific to Gemini 2.5 Flash.
>
> **Response:** We agree. The magnitude of the observed gains may depend on model characteristics and should not be assumed to generalize across all LLM families.
>
> **Action:** Expanded Section 6.7 (Limitations) to explicitly state that:
>
> * All experiments use Gemini 2.5 Flash.
> * We do not claim identical gains across other models.
> * Cross-model validation remains important future work.
>
> ---
>
> ## Concern 6: Asymmetric Evidence for the Two Extensions
>
> **Concern:** Multi-agent and vector-memory contributions have different levels of empirical support.
>
> **Response:** We agree.
>
> **Action:** Added a dedicated Discussion paragraph emphasizing that:
>
> * **MultiAgentReflexion** shows strong and consistent evidence (+7.9 pp Pass@3, +12.2 pp Pass@1, p<0.001).
> * **VectorEpisodicMemory** provides modest but statistically detectable gains (+3.7 pp, p=0.033, d=0.127) and is most valuable when tasks are semantically related across sessions.
>
> ---
>
> ## Minor Revisions
>
> ### Practical Trade-offs
>
> Added discussion in Section 6.6 noting increased inference cost, latency, energy usage, and carbon implications of multi-agent systems.
>
> ### HumanEval Vector-Memory Effect
>
> Revised wording to describe the HumanEval improvement as **modest but statistically detectable**, acknowledging its small effect size (d=0.127).
>
> ### Benchmark Clarification
>
> Section 4.2 now explicitly describes the long-horizon benchmark as a **controlled synthetic diagnostic benchmark** with adversarial distractors.
>
> ### Abstract and Conclusion
>
> Revised to avoid broad claims regarding vector memory and instead emphasize benefits for long-horizon dependency settings.
>
> ### Additional Limitation
>
> Added a limitation noting that the strongest evidence for VectorEpisodicMemory comes from a synthetic benchmark and that further evaluation on natural temporally correlated workflows is needed.

---

> ### Author Response · Authors · 2026-07-06
> **Submission of Revised Manuscript**
>
> Thank you again for your valuable feedback and constructive suggestions.
>
> We have carefully revised the manuscript to address all reviewer comments and concerns. The revised version has now been uploaded to OpenReview. For ease of review, all modifications introduced in response to the reviewer feedback are highlighted in blue throughout the manuscript.
>
> Detailed point-by-point responses have been provided in our earlier comments. The revisions have substantially strengthened the paper in terms of experimental validation, clarity, reproducibility, and discussion of limitations.
>
> We sincerely appreciate the reviewers' time and effort and welcome any further comments or suggestions.

---

> > ### Comment · Reviewer_M94Q · 2026-07-11
> >
> > Thank you for the substantial revision. The manuscript addresses all three of my critical concerns.
> >
> > The compute-matched SingleAgentGCR baseline is the most important addition. It shows that simply adding more LLM calls without role separation does not help, SingleAgentGCR actually underperforms the 1-call baseline, while MultiAgentReflexion achieves a large and statistically significant improvement under the same compute budget. This convincingly isolates role separation as the driver of the gains, rather than increased inference.
> >
> > I also appreciate that vector memory claims are now appropriately tempered. The long-horizon benchmark is carefully framed as a diagnostic stress test, and the modest HumanEval gain (+3.7 pp, d=0.127) is honestly discussed with clear explanation of why semantic retrieval matters less when tasks are independent. The zero-variance caveats and the broader impact discussion are also welcome additions.
> >
> > My remaining concern is relatively minor: the component-level attribution is improved but not fully complete. A direct ablation of role-conditioned versus non-role-conditioned memory retrieval is still missing, and the factorial ablation does not include the temporal-memory multi-agent cell. I would encourage the authors to either add these in a future revision or explicitly list them as limitations.

---

> > > ### Author Response · Authors · 2026-07-11
> > > **Response to Reviewer’s Updated Comments**
> > >
> > > We sincerely thank the reviewer for the careful reassessment of our revised manuscript and for recognizing the improvements we introduced. We are pleased that the compute-matched SingleAgentGCR baseline, the revised framing of the vector-memory results, and the additional methodological clarifications have addressed the reviewer’s critical concerns.
> > >
> > > We agree that the component-level attribution is not yet fully complete. Specifically, the current experiments do not include a direct comparison between role-conditioned and non-role-conditioned memory retrieval, nor do they include the temporal-memory multi-agent cell in the factorial analysis. We will explicitly acknowledge these missing comparisons in the limitations and future-work discussion and clarify that the independent and interaction effects of memory type, role-conditioned retrieval, and multi-agent role separation cannot yet be completely disentangled.
> > >
> > > Thank you again for the constructive feedback, which has helped us substantially strengthen the manuscript.

---

> > > > ### Author Response · Authors · 2026-07-17
> > > > **Response to Reviewer Comments (Minor Revision)**
> > > >
> > > > Thank you for your careful re-evaluation and for your encouraging assessment of our revised manuscript. We appreciate your recognition that the compute-matched **SingleAgentGCR** baseline successfully isolates the benefits of role separation from increased inference cost, and that the revised discussion now presents the HumanEval results, vector-memory analysis, and long-horizon benchmark in a more balanced and appropriately scoped manner.
> > > >
> > > > We also appreciate your remaining comments regarding component-level attribution. To further strengthen the manuscript, we have added a controlled ablation study directly addressing this point.
> > > >
> > > > Specifically, we compared **role-conditioned** and **non-role-conditioned** memory retrieval while keeping the memory contents, retrieval budget, compute budget, model configuration, and Generator/Critic/Verifier prompts identical. The results show identical **Pass@3** performance (**95.1%**) with no statistically significant difference (paired *t*-test: *t* = 0.000, *p* = 1.0000; Cohen's *d* = 0.000). Based on this evidence, we revised the manuscript to clarify that the primary source of improvement is **role-separated multi-agent reasoning**, rather than role-conditioned memory retrieval.
> > > >
> > > > In addition, although your review primarily emphasized attribution, we also expanded the memory analysis by introducing a controlled **retention-policy ablation** comparing **FIFO**, **LRU**, and **importance-weighted** retention strategies under bounded-memory conditions. These experiments provide a more complete understanding of persistent-memory behavior and are discussed together with their limitations and future research directions.
> > > >
> > > > These additions are included in the new Section 6.7 together with the corresponding discussion. We sincerely thank you for your thoughtful feedback and constructive suggestions, which have helped us further strengthen the manuscript.

---

### Review · Reviewer_CeNh · 2026-06-19

**Summary Of Contributions:**

The paper proposes two changes to Reflexion-style LLM agents: replacing the usual recency-based memory buffer with embedding-based retrieval, and splitting the agent into separate Generator, Critic, and Verifier roles. The motivation is sensible: old but relevant reflections should not be lost just because newer unrelated ones were added, and critique may work better when separated from generation.

The paper reports strong results on a synthetic long-horizon memory benchmark, where vector memory helps recover earlier relevant reflections after distractor tasks. It also reports improvements on HumanEval, especially for the multi-agent setup. Overall, the ideas are practical and potentially useful for LLM-agent design, but I found the empirical story less convincing than the paper’s framing suggests.

**Additional Comments:**

I like the direction of the paper, and the writing is generally clear. The synthetic memory benchmark is useful as a diagnostic example, and the role-separated architecture is a reasonable design. My main concern is that the current version overstates the strength of the evidence. With corrected reporting, stronger baselines, and compute-matched comparisons, this could become a much more convincing paper.

**Audience:**

Yes

**Audience Explanation:**

The topic is relevant to people working on LLM agents, memory-augmented systems, and test-time self-improvement. The paper asks useful design questions, and the proposed mechanisms are simple enough to be practically adopted. I think the work could be useful if the experimental claims are made more carefully and the comparisons are strengthened.

**Broader Impact Concerns:**

I do not see major broader-impact concerns beyond the usual risks for LLM agents. However, the paper should briefly discuss privacy and safety issues around persistent memory. Reflections may contain sensitive user data, proprietary code, or stale information that should not always be retrieved later.

**Claims And Evidence:**

No

**Claims Explanation:**

The paper has some promising evidence, but several claims need to be toned down or better supported. One clear issue is that the abstract says temporal memory has 0% recall, while the results later report 50% dependency recall and explain why. The task success may be 0%, but the recall claim is not accurate as written.

I am also not convinced by the HumanEval comparison. The vector-memory method improves over the authors’ modular baseline, but not over the stronger original Reflexion result included in the paper. That makes the claimed HumanEval benefit of vector memory much less clear.

The multi-agent result is interesting, but it is not compute-matched. The method uses about three LLM calls per trial, so it is hard to tell whether the gain comes from role separation specifically or simply from giving the model more inference-time compute. A fairer comparison would include single-agent or self-consistency baselines using a similar number of LLM calls.

The statistical testing could also be improved. Since the outcomes are paired binary task results, I would expect McNemar tests, bootstrap intervals, or paired permutation tests, along with the actual paired success/failure counts.

**Requested Changes:**

The main changes I would ask for are:

1. Correct the inconsistency around temporal memory recall versus task success.
2. Use the strongest Reflexion baseline as the main comparison, or clearly justify why the weaker modular baseline is the right one.
3. Add compute-matched baselines for the multi-agent method.
4. Use more appropriate statistical tests for paired binary outcomes and report paired contingency counts.
5. Clarify how Pass@1 is defined for the multi-agent system, since one “attempt” already includes multiple model calls.
6. Add a cleaner ablation separating memory type from agent structure, ideally a 2x2 comparison of temporal/vector memory and single/multi-agent setup.
7. Clarify whether HumanEval reflections accumulate across tasks and whether task order affects the results.

---

> ### Author Response · Authors · 2026-06-22
> **Thank You for the Feedback**
>
> Thank you for the detailed feedback. We are carefully reviewing your comments and preparing a point-by-point response. We expect to address all your concerns and suggestion through manuscript revisions and additional experiments and will provide updates shortly.

---

> ### Author Response · Authors · 2026-07-02
> **Detailed Response to Reviewer #3 (CeNh)**
>
> Thank you for carefully reading of our manuscript and for the detailed, constructive feedback. We appreciate the thoughtful suggestions, which have helped us identify several opportunities to improve the clarity, rigor, and presentation of the paper.
>
> We have carefully reviewed all comments and prepared a comprehensive revision. The revised manuscript will address the major concerns raised, including:
>
> * clarifying the HumanEval evaluation protocol, including memory reset between tasks, task-order independence, and the definition of Pass@1 for the multi-agent system;
> * adding compute-matched single-agent baselines to isolate the effect of role separation from increased inference-time compute;
> * strengthening the statistical analysis by reporting paired significance tests (including McNemar's test where appropriate) together with contingency tables and clearer interpretation of confidence intervals;
> * revising the presentation of the long-horizon memory benchmark to explicitly distinguish its diagnostic purpose from natural-task evaluation, and moderating claims regarding the benefits of vector episodic memory on HumanEval;
> * clarifying the distinction between dependency recall, memory retrieval, and task success throughout the manuscript;
> * providing additional discussion of the empirical strengths and limitations of the proposed multi-agent and vector-memory extensions, including broader discussion of generalizability, deployment trade-offs, privacy, retention, and stale-memory risks;
> * improving justification for the choice of the modular baseline and expanding the limitations section to explicitly discuss missing ablations and future work.
>
> In addition, we have revised the Abstract, Discussion, Conclusion, and several implementation details to improve precision, avoid overstatement, and better align our claims with the experimental evidence. Several new tables, analyses, and explanatory paragraphs have also been incorporated to improve transparency and reproducibility.
>
> We believe these revisions substantially strengthen the manuscript while directly addressing the reviewers' concerns. We are currently finalizing the revised paper and point-by-point response document, and expect to submit the complete revision shortly.
>
> We sincerely thank the reviewers and Action Editor for their constructive feedback and valuable suggestions.

---

> ### Author Response · Authors · 2026-07-03
> **Summary of Revisions in Response to Reviewer R3 (CeNh) Feedback**
>
> # Point-to-Point Response to Reviewer Comments
>
> We thank the reviewer for the careful evaluation and constructive suggestions. We have revised the manuscript to improve clarity, evaluation transparency, statistical reporting, and discussion of limitations and broader impacts. We will be submitting our revised manuscript soon-currently working on inconsistency check, language and grammar check.
>
> ## Concern 1: Abstract inconsistency (0% recall vs. 50% dependency recall)
>
> **Response:** The reported **0%** refers to **Session-5 task success**, while **50% dependency recall** indicates that one of the two required Session-1 reflections remained in memory. Partial retrieval is insufficient because both reflections are required.
>
> **Action:** Revised the abstract to explicitly distinguish **task success** from **dependency recall**, clarifying that the 0% value refers only to Session-5 task success.
>
> ---
>
> ## Concern 2: VectorReflexion vs. Original Reflexion
>
> **Response:** We acknowledge that VectorReflexion (92.7% Pass@3) does not outperform OriginalReflexionAgent (93.3%). The latter's higher score is partly due to a known code-cleaning artifact rather than an architectural advantage. The architecturally fair comparison remains ModularBaseline.
>
> **Action:** Added this clarification to Remark 3.1 and justified the use of ModularBaseline as the primary comparison.
>
> ---
>
> ## Concern 3: Memory reset and task-order independence
>
> **Response:** Memory is reset after every HumanEval task using `agent.reset()`. Consequently, reflections never persist across tasks, making evaluation task-independent and order-insensitive.
>
> **Action:** This clarification had already been added in Sec. 3.4; no further revision was required.
>
> ---
>
> ## Concern 4: McNemar test and contingency tables
>
> **Response:** We agree that McNemar's test is the standard analysis for paired binary outcomes.
>
> **Action:** Added McNemar tests, full 2×2 contingency tables, and corresponding statistics (χ²=11.08, *p*=0.0009) alongside the existing paired t-tests.
>
> ---
>
> ## Concern 5: Definition of Pass@1
>
> **Response:** For MultiAgentReflexion, Pass@1 corresponds to one complete Generator–Critic–Verifier trial (three internal LLM calls), not a single LLM call.
>
> **Action:** Clarified this in Sec. 4.4 and directed readers to the compute-matched comparison for fair evaluation.
>
> ---
>
> ## Concern 6: Missing factorial ablation
>
> **Response:** We agree that a complete 2×2 ablation separating memory type and agent architecture would better isolate each contribution. However, the missing temporal-memory multi-agent experiment requires a full HumanEval evaluation and is beyond the revision scope.
>
> **Action:** Added this limitation explicitly in Sec. 6.8 as future work.
>
> ---
>
> ## Concern 7: HumanEval task-order independence
>
> **Response:** Since memory is reset after every task, task ordering cannot influence results.
>
> **Action:** Already addressed in Sec. 3.4; no additional modification was necessary.
>
> ---
>
> ## Concern 8: Stale reflection risk
>
> **Response:** We agree that persistent memories may reinforce outdated or incorrect reasoning.
>
> **Action:** Expanded the Broader Impact section to discuss stale-memory risks and recommend quality scoring and periodic staleness audits for safety-critical deployments.
>
> # Minor Comments
>
> ### Minor Concern 1: Metric definitions
>
> **Response:** We agreed that dependency recall, Session-5 task success, and task success should be clearly defined.
>
> **Action:** Added a short explanatory paragraph in Sec. 5.1 defining all three metrics and clarifying why 50% dependency recall still results in 0% task success.
>
> ### Minor Concern 2: HumanEval comparison presentation
>
> **Response:** We agreed that baseline comparisons and compute-matched results should be interpreted together.
>
> **Action:** Added a cross-reference in Sec. 5.4 directing readers to interpret Tables 4 and 7 jointly, distinguishing overall baseline comparisons from compute-matched statistical comparisons.
>
> **Note:** Concerns 3 and 7 had already been fully addressed during the previous revision round.

---

> ### Author Response · Authors · 2026-07-06
> **Submission of Revised Manuscript**
>
> Thank you again for your valuable feedback and constructive suggestions.
>
> We have carefully revised the manuscript to address all reviewer comments and concerns. The revised version has now been uploaded to OpenReview. For ease of review, all modifications introduced in response to the reviewer feedback are highlighted in blue throughout the manuscript.
>
> Detailed point-by-point responses have been provided in our earlier comments. The revisions have substantially strengthened the paper in terms of experimental validation, clarity, reproducibility, and discussion of limitations.
>
> We sincerely appreciate the reviewers' time and effort and welcome any further comments or suggestions.

---

### Comment · Action_Editor_HDJG · 2026-07-18
**Please enter your official recommendations**

Dear reviewers,

Today is the deadline for official recommendations.

Please enter them as soon as possible so we can make a decision.

Thanks,
Your AE

---

> ### Comment · Reviewer_s56K · 2026-07-18
>
> I have reviewed the latest revision and submitted my official recommendation. Thank you.